# A New Method for Calculating Number Concentrations of Cloud Condensation Nuclei Based on Measurements of A Three-wavelength Humidified Nephelometer System

Jiangchuan Tao[1], Chunsheng Zhao[1], Ye Kuang[1], Gang Zhao[1], Chuanyang Shen[1], Yingli Yu[1], Yuxuan Bian[2], Wanyun Xu[2]

[1]{Department of Atmospheric and Oceanic Sciences, School of Physics, Peking University, Beijing, China}

[2]{State Key Laboratory of Severe Weather, Chinese Academy of Meteorological Sciences}

*Correspondence to: C. S. Zhao (zcs@pku.edu.cn)

Abstract

The number concentration of cloud condensation nuclei (CCN) plays a fundamental role in cloud physics. Instrumentations of direct measurements of CCN number concentration ($N_{CCN}$) based on chamber technology are complex and costly, thus a simple way for measuring $N_{CCN}$ is needed. In this study, a new method for $N_{CCN}$ calculation based on measurements of a three-wavelength humidified nephelometer system is proposed. A three-wavelength humidified nephelometer system can measure aerosol light scattering coefficient ($\sigma_{sp}$) at three wavelengths and the light scattering enhancement factor (fRH). The Angstrom exponent (Å) inferred from $\sigma_{sp}$ at three wavelengths provides information on mean predominate aerosol size and hygroscopicity parameter ($\kappa$) can be calculated from the combination of fRH and Å. Given this, a look-up table that includes $\sigma_{sp}$, $\kappa$ and Å is established to predict $N_{CCN}$. Due to the precondition for the application, this new method is not suitable for externally mixed particles, large particles (e.g. dust and sea salt) or fresh aerosol particles. This method is validated with direct measurements of $N_{CCN}$ using a CCN counter on the North China Plain. Results show that relative deviations between calculated $N_{CCN}$ and measured $N_{CCN}$ are within 30% and confirm the robustness of this method. This method enables simpler $N_{CCN}$ measurements because the humidified nephelometer system is easily operated and stable. Compared with the

method of CCN counter, another advantage of this newly proposed method is that it can obtain $N_{CCN}$
at lower supersaturations in the ambient atmosphere.

1.  Introduction
Cloud condensation nuclei (CCN) are the aerosol particles forming cloud droplet by hygroscopic
growth. CCN number concentration ($N_{CCN}$) plays a fundamental role in cloud microphysics and
aerosol indirect radiative effect. In general, the direct measurement of $N_{CCN}$ is achieved in a chamber
under super-saturated conditions (Hudson, 1989;Nenes et al., 2001;Rose et al., 2008). Due to the
requirement of high accuracies of working conditions like temperatures, vapors and flow rates in
chambers, the direct measurement of $N_{CCN}$ is complex and costly (Rose et al., 2008;Lathem and
Nenes, 2011). Thus, developments of simplified measurements of $N_{CCN}$ are required. In recent years,
attention has been focused on measurements of aerosol optical properties (Jefferson, 2010;Ervens et
al., 2007;Gasso and Hegg, 2003), which are simple and well-developed (Covert et al., 1972;Titos et
al., 2016). For aerosol population free of sea salt or dust, the accumulation mode aerosol not only
dominates aerosol scattering ability but also contribute most to $N_{CCN}$. Thus, the calculation of $N_{CCN}$
based on measurements of aerosol optical properties is feasible, and can facilitate $N_{CCN}$
measurement.
There are two kinds of methods to calculating $N_{CCN}$ based on measurements of aerosol optical
properties. For the first kind, $N_{CCN}$ as well as the hygroscopicity parameter ($\kappa$) can be calculated
based on measurements of a humidified nephelometer system in combination with aerosol particle
number size distribution (PNSD) (Ervens et al., 2007;Chen et al., 2014). Thus additional
measurements of PNSD are needed. For the second kind, $N_{CCN}$ is calculated based on statistical
relationships between $N_{CCN}$ and aerosol optical properties, such as scattering coefficient ($\sigma_{sp}$),
Angstrom Exponent (Å and single scattering albedo (SSA) (Jefferson, 2010;Shinozuka et al., 2015).
Å is the exponent commonly used to describe the dependence of $\sigma_{sp}$ on wavelength as the formula
shows:
$$\sigma_{sp}(\lambda)=\beta \cdot \lambda^{-Å}, \tag{1}$$
where $\beta$ is the aerosol number concentration. Coefficient of determination ($R^2$) between measured

and calculated $N_{CCN}$ using the first kind of method is about 0.9. For the second kind of method, $R^2$ is generally lower than 0.9, although the used instruments are cheaper and easier in operation. Applications similar to the second kind are widely used in remote sensing. As shown in Table 1, earlier studies found that the aerosol volume or aerosol PNSD retrieved from remote sensing measurements can be used to calculate $N_{CCN}$ (Gasso and Hegg, 2003;Kapustin et al., 2006). Recently, either aerosol optical depth (AOD) or aerosol vertical profile is used to predict $N_{CCN}$ directly(Ghan and Collins, 2004;Ghan et al., 2006;Andreae, 2009;Liu and Li, 2014).

In the statistical relationship between $N_{CCN}$ and aerosol optical properties, $\sigma_{sp}$ or AOD is mainly the proxy of aerosol absolute concentration, while Å or SSA can be used to reveal the variations of aerosol CCN activity, as shown in Table 1. Based on Kohler theory (Köhler, 1936;Petters and Kreidenweis, 2007), aerosol CCN activity is determined by aerosol size and aerosol chemical composition, and aerosol chemical composition can be defined as aerosol hygroscopicity. Information about aerosol size and aerosol hygroscopicity are critical to $N_{CCN}$ prediction and their absence can lead to a deviation with factor of four (Andreae, 2009). Compared with aerosol hygroscopicity, aerosol size is more important in determining CCN activity (Dusek et al., 2006). The value of Å can provide information on mean predominate aerosol size (Brock et al., 2016;Kuang et al., 2017a). As a result, $N_{CCN}$ calculation from Å and extinction coefficient is found to be accurate to some extent (Shinozuka et al., 2015). As proxies for aerosol hygroscopicity, SSA or aerosol light scattering enhancement factor (fRH) is commonly used while not so effective (Jefferson, 2010; Liu and Li, 2014). fRH is defined as:

$$fRH=\sigma_{sp}(RH)/\sigma_{sp} \qquad (2)$$

where $\sigma_{sp}(RH)$ is the humidified $\sigma_{sp}$ at a given RH. SSA is determined by the ratio between the light absorbing carbonaceous and less-absorbing components. Black carbon dominates the absorption of solar radiation and is a main hydrophobic components as well. Less-absorbing components consist of inorganic salts and acids, as well as most organic compounds, which are generally hygroscopic components. SSA correlates positively with aerosol hygroscopicity (Rose et al., 2010) but deviates significantly due to the diversity of hygroscopicity of less-absorbing components. Thus $N_{CCN}$ calculation combining SSA, backscatter fraction and $\sigma_{sp}$ still leads to

significant deviations, with $R^2$ = 0.6 (Jefferson, 2010). As for fRH, there was a study that applied aerosol optical quantities ($\sigma_{sp}$ or aerosol optical thickness) with fRH or SSA to calculate $N_{CCN}$ (Liu and Li, 2014). In their study, compared with the combination of SSA and aerosol optical quantities, the combination of fRH and aerosol optical quantities is found to be less accurate in estimating $N_{CCN}$, even though fRH is directly connected with aerosol hygroscopicity (Liu and Li, 2014). This may result from the significant dependence of fRH on aerosol size(Chen et al., 2014;Kreidenweis and Asa-Awuku, 2014;Kuang et al., 2017a). As mentioned before, PNSD is used for better calculation of $\kappa$ and $N_{CCN}$ from fRH in previous studies (Ervens et al., 2007;Chen et al., 2014). A new method to estimate $\kappa$ from fRH and Å was proposed recently (Kuang et al., 2017a;Brock et al., 2016). Based on this method, fRH can be used to calculate $N_{CCN}$ without measurements of PNSD and can be expected to improve the $N_{CCN}$ prediction just based on measurements of aerosol optical properties.

In this study, the relationship between $N_{CCN}$ and aerosol optical properties measured by a humidified nephelometer system is studied and a new method for $N_{CCN}$ prediction is proposed. This new method is validated based on data observed in Gucheng campaign on the North China Plain and can be expected to improve measurements of $N_{CCN}$ due to advantages of applying nephelometers.

## 2. Methodology

### 2.1. Data

Data in this study are mainly measured at Gucheng (39.15N, 115.74E) during autumn in 2016 on the North China Plain (NCP). Gucheng is 100km southwest from Beijing and 40km northeast from Baoding under background pollution condition in the NCP. The observation site was surrounded by farmland and about 3km away from the Gucheng town. This campaign started on 20 October and lasted for nearly one month.

Instruments used in Gucheng campaign were located in a measurement container under temperature maintained at 25 ∘C. Ambient aerosol was sampled and dried to relative humidity (RH) lower than 30% by an inlet system consisting of a PM10 inlet, an inline Nafion dryers and a RH and temperature sensor (Vaisala HMP110). Then the sample aerosol was separated by a splitter and directed into various instruments. During this campaign, $\sigma_{sp}$, fRH, particle size-resolved activation

ratio (AR) and particle number size distribution (PNSD) were obtained.

111        fRH as well as $\sigma_{sp}$ at three wavelengths were measured by a humidified nephelometer system

consisting of two nephelometers (Aurora 3000, Ecotech Inc.) and a humidifier. In addition, Å can be
calculated directly from $\sigma_{sp}$ measured by a nephelometer. The humidifier with a Gore-Tex tube
humidified the sample air up to 90% RH. A whole cycle of humidification lasted about 45minutes
from 50% RH to 90% RH. Dried $\sigma_{sp}$ was obtained directly from dried sample aerosol measured by
one nephelometer and humidified $\sigma_{sp}$ was obtained from humidified aerosol measured by another
nephelometer. fRH can be calculated by dividing humidified $\sigma_{sp}$ by dried $\sigma_{sp}$. Detailed description
of the humidified nephelometer system was illustrated in Kuang et al (2017a).

119        The particle size-resolved activation ratio (AR), defined as the ratio of $N_{CCN}$ to total particles,

was measured by a system mainly consisting of a differential mobility analyzer (DMA, Model 3081)
and a continuous-flow CCN counter (model CCN200, Droplet Measurement Technologies, USA;
Roberts and Nenes (2005); Lance et al., (2006)). The system selected mono-disperse particles with
the DMA coupled with an electrostatic classifier (model 3080; TSI, Inc., Shoreview, MN USA) and
measured AR of the mono-disperse particles by a condensation particle counter (CPC model 3776;
TSI, Inc.) and CCN counter. Ranges of particle size and supersaturation were 10-300nm and
0.07%-0.80%, respectively. Measurements at five supersaturations (0.07%, 0.10%, 0.20%, 0.40%
and 0.80%) were conducted sequentially with each cycle lasted for 1 hour, and $N_{CCN}$ at 0.07%
supersaturation was used in this study. Due to non-idealities of CCN counter at supersaturations
lower than 0.10%, CCN measurement at 0.07% supersaturation was found to be the most uncertain
(Rose et al., 2008) and can lead to deviations of measured $N_{CCN}$ in this study. Before and after the
campaign, supersaturations set in this system were calibrated using ammonium sulfate (Rose et al.,
2008). More information about the system is available in Deng et al. (2011) and Ma et al.(2016).

133        PNSD with particle diameter from 9nm to 10um was measured by a mobility particle size

spectrometer (SMPS, TSI Inc., Model 3996) and an Aerodynamic Particle Sizer (APS, TSI Inc.,
Model 3321). SMPS consisted of a DMA, an electrostatic classifier and a CPC (model 3776; TSI,
Inc., Shoreview, MN USA) and measured PNSD with diameter lower than 700nm.

137        In addition, PNSD and $\sigma_{sp}$ from 2011 to 2014 at four campaigns (Wuqing in 2011, Xianghe in

2012 and 2013, and Wangdu in 2014) in NCP were used in this study. PNSD in these campaigns was
measured by a Twin Differential Mobility Particle Sizer (TDMPS, Leibniz-Institute for Tropospheric
Research (IfT), Germany) and an Aerodynamic Particle Sizer (APS, TSI Inc., Model 3321). A TSI
3563 nephelometer was used to obtain $\sigma_{sp}$ at three wavelengths. Details about the four campaigns
can be found in Ma et al. (2011), Ma et al.(2016), Kuang et al. (2016) and Kuang et al.(2017a).

2.2.  Theories
Hygroscopic growth of particles at certain relative humidity can be described by $\kappa$-Köhler
theory (Petters and Kreidenweis, 2007):
$$\frac{RH}{100} = \frac{g(RH)^3 - 1}{g(RH)^3 - (1-\kappa)} \cdot \exp\left(\frac{4\sigma_{s/a} \cdot M_w}{R \cdot T \cdot D_d \cdot g(RH) \cdot \rho_w}\right) \qquad (3)$$
where g(RH) is geometric diameter growth factor, $\kappa$ is the hygroscopicity parameter, RH is the
relative humidity; $\rho_W$ is the density of water; $M_W$ is the molecular weight of water; $\sigma_{s/a}$ is the surface
tension of the solution–air interface, which is assumed to be equal to the surface tension of the pure
water–air interface; R is the universal gas constant; and T is the temperature.
Accounting for the impact of Å, $\kappa_f$ can be derived directly from fRH (Brock et al., 2016;Kuang
et al., 2017a). A single-parameter parameterization scheme proposed by Brock et al. (2016) connects
fRH and $\kappa$ by the approximately proportional relationship between total aerosol volume and $\sigma_{sp}$:
$$f(RH)=1+\kappa_{sca} *RH/(100-RH) \qquad (4)$$
where $\kappa_{sca}$ is a parameter for fitting fRH curves and is found can be used to predict $\kappa_f$ in
combination with Å in recent studies (Brock et al., 2016;Kuang et al., 2017a). This method of
calculating $\kappa_f$ based on $\kappa_{sca}$ and Å was confirmed by good agreement with $\kappa_f$ calculated from
fRH and PNSD.
$N_{CCN}$ can be calculated from size-resolved AR at a certain supersaturation (SS) and PNSD
(referred to as n(log$D_p$)) as follows:
$$N_{CCN}= \int_{\log D_P} AR(\log D_P ,SS) \cdot n(\log D_P )d\log D_P \qquad (5)$$
In general, size-resolved AR curves are complicated and always replaced by a critical diameter ($D_c$)
to simplify calculation (Deng et al., 2013). The critical diameter is defined as:
$$N_{CCN} = \int_{\log D_c}^{\log D_{P,max}} n(\log D_P)\, \mathrm{d}\log D_P \qquad (6)$$

where $D_{P,max}$ is the maximum diameter of the measured particle number size distribution. In other
words, the integral of PNSD larger than $D_c$ equals to the measured $N_{CCN}$. And a critical $\kappa$ ($\kappa_c$) can be
calculated by equation (3) and indicates CCN activity and hygroscopicity of particles.

3.  Results
3.1.  Calculation of $N_{CCN}$ based on measurements of a Humidified Nephelometer system
Free of sea salt aerosol and dust aerosol, accumulation mode aerosol dominates both the optical
scattering ability at short wavelengths and the CCN activity at low supersaturations, and thus a
reasonable relationship between $\sigma_{sp}$ and $N_{CCN}$ can be achieved. Figure 1 shows the size distribution
of cumulative contributions of $\sigma_{sp}$ at 450nm and $N_{CCN}$ at 0.07% with various Å and $\kappa_c$, and
corresponding normalized PNSDs based on data measured at the four campaigns on the North China
Plain. During the four campaigns, no sea salt aerosol or dust aerosol was observed(Ma et al.,
2011;Ma et al., 2016;Kuang et al., 2016;Kuang et al., 2017a). For continental aerosol without sea salt
or dust, Å varies from 0.5 to 1.8 and $\kappa_c$ varies from 0.1 to 0.5 (Cheng et al., 2008;Ma et al.,
2011;Liu et al., 2014;Kuang et al., 2017b). And as mentioned before, Å can be used as a proxy of
the overall size distribution of aerosol populations, with smaller Å indicating more larger particles.
In figure 1, comparisons for Å are made between 0.5 and 1.9 and for $\kappa_c$ are made between 0.1 and
0.5. As larger particles contribute more to light scattering and CCN activation, cumulative
contributions of both $\sigma_{sp}$ and $N_{CCN}$ increase significantly at the diameter range of accumulation
mode particles. Because more hygroscopic particles are able to activate at smaller diameters, the
cumulative contribution of $N_{CCN}$ with higher $\kappa_c$ increases at smaller diameters. In general, major
contributions of both $\sigma_{sp}$ and $N_{CCN}$ are made by particles from 200nm to 500nm for various Å and
$\kappa_c$. This implies the feasibility of inferring $N_{CCN}$ from aerosol optical properties.
Because particles smaller than 200nm can activate at supersaturations higher than 0.07% while
scatter less light at wavelengths longer than 450nm, which are shown as the light color lines in
Figure 1, it's obvious that significant differences will exist between cumulative contributions of $\sigma_{sp}$
and $N_{CCN}$. This means $\sigma_{sp}$ and $N_{CCN}$ are dominated by different particles and poor correlation
between $\sigma_{sp}$ and $N_{CCN}$ can be expected. Thus the method of inferring $N_{CCN}$ from aerosol optical
properties is applicable for shorter wavelength and lower supersaturations.
Furthermore, PNSD with higher Å indicates more Aitken mode particles and fewer
accumulation mode particles. Thus large particles contribute less for both $\sigma_{sp}$ and $N_{CCN}$ when Å are
higher, characterizing an increase of cumulative contribution curves at smaller diameters. In detail,
cumulative contribution curves of $\sigma_{sp}$ at 1.9 Å is about 0.3 higher than these curves at 0.5 Å at the
size range of 200nm to 700nm. While cumulative contribution curves of $N_{CCN}$ at 1.9 Å is no higher
than 0.2 higher than these curves at 0.5 Å. Changes of cumulative contributions of $N_{CCN}$ and $\sigma_{sp}$
with various Å reveal that the shape of PNSD can influence the correlation between $N_{CCN}$ and $\sigma_{sp}$.
This is confirmed by previous studies in which the Å is found to play an important role in
calculating $N_{CCN}$ from $\sigma_{sp}$ (Shinozuka et al., 2015;Liu and Li, 2014).
The relationship between $\sigma_{sp}$ and $N_{CCN}$ dependent on Å and $\kappa_c$ is evaluated by calculating
$\sigma_{sp}$ and $N_{CCN}$ with different PNSDs (classified by Å) and different $\kappa_c$. In detail, ratios of $N_{CCN}$ to
$\sigma_{sp}$, referred to as $AR_{sp}$, are calculated to eliminate the effect of variations of particle concentrations
consistent at all diameters. Results at the supersaturation of 0.07% are shown in figure 2 and $AR_{sp}$ is
higher than 0 and lower than 10. In general, $AR_{sp}$ are higher for more hygroscopic particles or
smaller particles. As particles become more hygroscopic, more CCN can be expected when $\sigma_{sp}$ is
fixed. As aerosol populations consist of more smaller CCN-active particles, the increase of $\sigma_{sp}$ is
weaker than that of $N_{CCN}$. For example, particles with diameters slightly larger than $D_c$ contribute
less to $\sigma_{sp}$ than particles with diameters much larger than $D_c$.
In detail, the sensitivity of $AR_{sp}$ to $Å$ also changes with $Å$ and $\kappa_c$. When $Å$ are higher than 1.4
and $\kappa_c$ is lower than 0.2, $AR_{sp}$ is insensitive to $Å$. While when $Å$ are lower than 1 and $\kappa_c$ are
higher than about 0.3, $AR_{sp}$ is more sensitive to $Å$ than $\kappa_c$. This higher sensitivity of $AR_{sp}$ to
$Å$reveals that, if the mean predominate size of particles is smaller, the increase of $N_{CCN}$ due to the
increase of $Å$ mentioned in the former paragraph can be larger as a result. This is the consequence
of the sensitivity of $AR_{sp}$ to $Å$ resulting from the variation of small CCN-active particles, as
mentioned before.
Based on the lookup-table illustrated in Figure 2, $N_{CCN}$ at the supersaturation of 0.07% can be
calculated simply from $Å$, $\kappa_f$ and $\sigma_{sp}$ which can be obtained from measurements of a humidified
nephelometer system. The description of this simple method is shown in figure 3. A new look-up
table needs to be made for $N_{CCN}$ estimation at other supersaturations, which should better be less than
0.07% as mentioned in the discussion of figure 1.
One critical issue about the method is the conversion of the $\kappa_f$ obtained from the humidified
nephelometer system to the $\kappa_c$ under super-saturated conditions. There are mainly two factors
making this conversion necessary. First, closure studies of aerosol hygroscopicity found significant
deviations between hygroscopicity at sub-saturated conditions and super-saturated conditions (Wex
et al., 2009; Irwin et al., 2010; Good et al., 2010; Renbaum-Wolff et al. 2016). Their difference can
be expected to be about 0.1 for accumulation mode aerosol(Wu et al., 2013;Whitehead et al.,
2014;Ma et al., 2016). Second, $\kappa_f$ indicates the hygroscopicity of total particles and can be quite
different from aerosol hygroscopicity at a specific diameter due to variations of size-dependent
particle hygroscopicity. Kuang et al. (2017a) found a difference around 0.1 between $\kappa_f$ and $\kappa$
inferred from g(RH) measurements for accumulation mode particles whose $\kappa_f$ is no larger than 0.2.
In this study, a simple conversion that $\kappa_c$ is 0.2 higher than $\kappa_f$ is used to calculate $N_{CCN}$, while for
$\kappa_f$ larger than 0.2, a smaller difference of 0.1 between $\kappa_c$ and $\kappa_f$ should be used (Kuang et al.,
2017a). This simplified relationship between $\kappa_c$ and $\kappa_f$ is a rough estimate regardless of the
complexity of differences of aerosol hygroscopicity measured by different instruments, but still used
in this study for two reasons. First, the accurate conversion cannot be achieved without detailed
information of the particle hygroscopicity, which is difficult and complicated to measure. Second, a
deviation of $\kappa_c$ less than 0.1 generally leads to a deviation of $N_{CCN}$ less than 20% (Ma et al., 2016),

which is comparable with the deviation of CCN measurements. As a result, for a simple method of $N_{CCN}$ calculation, this simple conversion is applicable. In addition, it is important to note that the value of the difference between $\kappa_c$ and $\kappa_f$ is also a rough estimate regardless of the complexity of aerosol hygroscopicity under different conditions, and the influence of $\Delta\kappa$ deviation on $N_{CCN}$ calculation needs to be further examined based on field observation. For fresh aerosol, the actual $\Delta\kappa$ can be too large (about 4 times of kappa values for some organic components, Wex et al., 2009; Renbaum-Wolff et al., 2016) or too small (nearly zero for inorganic components and black carbon) and thus is not suitable for the application of this method.

Besides aerosol size and hygroscopicity, aerosol mixing state can also affect aerosol CCN activity. When primary aerosol emissions are strong, aerosol populations are likely to be externally mixed and a realistic treatment of aerosol mixing state is critical for $N_{CCN}$ calculation (Cubison et al., 2008;Wex et al., 2010). But for regions away from strong aerosol primary emissions, the influence of mixing state on aerosol CCN activity is small and the assumption of internal mixing state is effective for the estimation of $N_{CCN}$ (Dusek et al., 2006;Deng et al., 2013;Ervens et al., 2010). For regions above the boundary layer where clouds form and measurements of $N_{CCN}$ are important, aerosol generally tends to be internally mixed when there is no strong vertical transport (McMeeking et al., 2011; Ferrero et al., 2014) and no plumes(Moteki and Kondo, 2007;McMeeking et al., 2011). In addition, it should be noted that influences of aerosol hygroscopicity and aerosol size on aerosol CCN activity are more significant than aerosol mixing state and the deviation of $N_{CCN}$ calculation due to the assumption of aerosol mixing state is smaller than the deviation due to aerosol size and aerosol hygroscopicity. In the new method of this paper, using Å and $\kappa_c$ to indicate the influence of aerosol size and aerosol hygroscopicity on aerosol CCN activity will increase the deviation of $N_{CCN}$ calculation, which is much larger than the deviation due to the assumption of aerosol mixing state. As a result, the improvement of $N_{CCN}$ calculation by introducing a more detailed mixing state than internal mixing is limited and aerosol populations are assumed to be internally mixed for simplification. Thus this method might not be applicable for regions or air masses greatly affected by strong primary aerosol emissions. Furthermore, this new method cannot be applied for regions where sea salt or dust prevails, as mentioned before. In summary, this method can be used to calculate $N_{CCN}$ for air mass tending to be dominated by aged aerosol particles like continental regions and clouds

forming heights.
3.2.   Validation based on $N_{CCN}$ measurement
The method for calculating $N_{CCN}$ based on measurement of the humidified nephelometer system,
including the conversion of $\kappa_c$ and the lookup-table, is examined using data measured in Gucheng.
Overview of data in Gucheng is shown in Figure 4. From polluted periods to clean periods,
significant variations of $N_{CCN}$ and $\sigma_{sp}$ can be found but $AR_{sp}$ of $N_{CCN}$ to $\sigma_{sp}$ stays around 5. On
October 23$^{rd}$ and 29$^{th}$, $N_{CCN}$ and $\sigma_{sp}$ are lower than 2000#/cm$^3$ and 500Mm$^{-1}$, respectively. While on
October 20$^{th}$, 26$^{th}$ and November 3$^{rd}$, $N_{CCN}$ and $\sigma_{sp}$ are higher than 2000#/cm$^3$ and 500Mm$^{-1}$,
respectively. These variations of $N_{CCN}$ and $\sigma_{sp}$ are mainly due to the variation of the particle number
concentration rather than the shape of particle size distribution and aerosol hygroscopicity. Variations
of $AR_{sp}$ result from the variations of Å and $\kappa_c$, which indicate the variations of aerosol
microphysical properties and chemical compositions.
In general, $AR_{sp}$ is more sensitive to variations of Å than $\kappa_c$. As mentioned before, the
sensitivity of $AR_{sp}$ to Å is determined by both Å and $\kappa_f$. In detail, Å during the campaign mainly
ranges from 0.5 to 1.5 and $\kappa_f$ ranges mainly from 0.05 to 0.2, which means that $\kappa_c$ ranges from
0.25 to 0.4. These values of Å and $\kappa_f$ correspond to a significant sensitivity of $AR_{sp}$ to Å, as the
lookup table shows in figure 2. The sensitivity of $AR_{sp}$ to $\kappa_c$ is much small and only notable during
some short periods (grey bars in Figure 4). For example, from November 5$^{th}$ to 7$^{th}$, variations of $\kappa_f$
and Å are opposite and result in nearly constant $AR_{sp}$. And from October 30$^{th}$ to November 2$^{nd}$,
consistent variations of Å and $\kappa_f$ lead to greater variations of $AR_{sp}$ than other periods. This weak
sensitivity of $AR_{sp}$ to $\kappa_f$ may be due to the uncertainty of $\kappa_c$ calculated from $\kappa_f$ based on the
simplified conversion.
Based on the lookup table of $\kappa_c$ and Å, $AR_{sp}$ is calculated and applied to calculate $N_{CCN}$ with
$\sigma_{sp}$. The calculated $AR_{sp}$ and $N_{CCN}$ are compared with the measured $AR_{sp}$ and $N_{CCN}$ shown as the
green dots in Figure 5. In general, good agreements between calculations and measurements are
achieved and relative deviations are within 30%. For the comparison of $AR_{sp}$, the system relative
deviation is less than 10%. For the comparison of $N_{CCN}$, the slope and the correlation coefficient of
the regression are 1.03 and 0.966, respectively.
In addition, the variation of $\Delta\kappa$ and its influence on $AR_{sp}$ and $N_{CCN}$ calculation are studied . As
shown in Figure 6, $\Delta\kappa$ is around 0.2 and independent from Å and $\kappa_c$ and over 80% of $\Delta\kappa$ ranges
from 0.1 to 0.3. A notable deviation of $\Delta\kappa$ can only be found when Å is higher than 1.5. High
values of Å represent existence of small particles, which tend to be fresh emitted and experience
inefficient aging processes. In this case, this simplified conversion of $\kappa_c$ may not be applicable.
Furthermore, $\Delta\kappa$ with different values are applied in the new method to calculate $N_{CCN}$. In the first
way, $\Delta\kappa$ of the $\kappa_c$ conversion is set to be 0.05 higher or lower, which means $\Delta\kappa$ of 0.25 or 0.15.
The corresponding results are presented as the red dots and blue dots in Figure 5. In the second way,
a constant $\kappa_c$ of 0.34, which is the average of $\kappa_c$ values in Gucheng campaign, is used to calculate
$AR_{sp}$ and $N_{CCN}$, and shown as the grey dots in Figure 5. In general, differences among calculations
using various $\kappa_c$ conversions are quite small. The $\Delta\kappa$ difference of 0.05 in $\kappa_c$ conversion only
leads to a difference of 10% for the system relative deviation of calculated $N_{CCN}$. The correlation
coefficient of the calculation using a constant $\kappa_c$ is just a little lower than correlation coefficients of
calculations using a $\kappa_c$ conversion. As a result, for data measured in Gucheng campaign, the method
of calculating $N_{CCN}$ is insensitive to the uncertainty of the $\kappa_c$ conversion and a $\Delta\kappa$ of 0.2 is
applicable in this new method.
In this study, the insensitivity of calculated $N_{CCN}$ to $\kappa_c$ conversion is partly due to the small
variation of $\kappa_f$ during the campaign. However, the variation of $\kappa_c$ can be quite large and cause
non-ignorable deviations of calculated $N_{CCN}$. As previous studies of $N_{CCN}$ measurement showed, the
variation of $\kappa_c$ is often small and a constant $\kappa_c$ can be used to calculate $N_{CCN}$ accurately (Andreae
and Rosenfeld, 2008;Gunthe et al., 2009;Rose et al., 2010;Deng et al., 2013). Results in this study
are similar to these previous studies. But large variations of $\kappa_c$ are also found in some other studies.
In NCP, fluctuations of aerosol hygroscopicity during New Particle Formation events and soot
emissions lead to significant deviations of calculated $N_{CCN}$ from average aerosol hygroscopicity (Ma
et al., 2016). Furthermore, the influence of $\kappa_c$ cannot be ignored because the value of the average
hygroscopicity is different in various regions during various periods. In summer of NCP, measured
$\kappa_f$ at sub-saturated conditions can reach up to 0.45 when inorganic components dominate in particles
(Kuang et al., 2016). In this case, calculated $N_{CCN}$ ignoring $\kappa_c$ may be 10 times larger than measured
$N_{CCN}$. To sum up, although the exact value of $\kappa_c$ cannot be obtained from the measurement of the
humidified nephelometer system, the influence of $\kappa_c$ on $N_{CCN}$ can be inferred and is found to be
correct enough considering the convenience of this method. More data, especially in observations of
more hygroscopic aerosol, is still needed to confirm this method.
4.  Conclusions
$N_{CCN}$ is a key parameter of cloud microphysics and aerosol indirect radiative effect. Direct
measurements of $N_{CCN}$ are generally conducted under super-saturated conditions in CCN chambers,
and are complex and costly. The aerosols of accumulation mode contribute most to both the aerosol
scattering coefficient and the aerosol CCN activity. In view of this, it is possible to predict $N_{CCN}$
based on relationships between aerosol optical properties and the aerosol CCN activity. In this study,
a new method is proposed to calculate $N_{CCN}$ based on measurements of a humidified nephelometer
system. In this method, $N_{CCN}$ is derived from a look-up table which involves $\sigma_{sp}$, Å and $\kappa_f$ , and
the required three parameters can be obtained from a three-wavelength humidified nephelometer
system.
Relationships between aerosol optical properties and aerosol CCN activity are investigated using
datasets about aerosol PNSD measured during several campaigns in the North China Plain. The
relationship between $\sigma_{sp}$, Å, $\kappa_c$ and $N_{CCN}$ is analyzed. It is found that the ratio between $N_{CCN}$ and
$\sigma_{sp}$, referred to as $AR_{sp}$, is determined by $\kappa_c$ and Å. In light of this, it is possible to calculate $N_{CCN}$
based only on measurements of a three-wavelength humidified nephelometer system which provides
information about $\sigma_{sp}$, the hygroscopicity parameter $\kappa$ and Å. However, $\kappa$ derived from
measurements of a humidified nephelometer system under sub-saturated conditions (termed as $\kappa_f$)
differs from $\kappa$ under super-saturated conditions which indicate CCN activity (termed as $\kappa_c$). As a
result, the conversion from $\kappa_f$ to $\kappa_c$ is needed. Based on previous studies of aerosol hygroscopicity
and CCN activity, a simple conversion from $\kappa_f$ to $\kappa_c$ with a fixed difference (referred to as $\Delta\kappa$ ) of
0.2 is proposed. On the basis of this simple conversion, the method of $N_{CCN}$ prediction based only on
measurements of a humidified nephelometer system is achieved under conditions without sea salt
aerosol, dust aerosol, externally mixed aerosol or fresh aerosol.
This method is validated with measurements of a humidified nephelometer system and a CCN
counter in Gucheng in 2016. During the campaign, both $N_{CCN}$ and $\sigma_{sp}$ vary with the pollution
conditions. $AR_{sp}$ is around 5 and changes with Å and $\kappa_f$. Based on this new method, $N_{CCN}$ are
calculated to compare with its measured values. The agreement between the calculated $N_{CCN}$ and the
measured $N_{CCN}$ is achieved with relative deviations less than 30%. Furthermore, the variation of $\Delta\kappa$
and its influence on $N_{CCN}$ calculation are studied. The difference between $\kappa_f$ and $\kappa_c$, was $0.2\pm0.1$.
Sensitivity of calculated $N_{CCN}$ to conversions from $\kappa_f$ to $\kappa_c$ is studied by applying different kinds of
conversions. Results show that calculated $N_{CCN}$ varies little and is insensitive to the conversions,
which confirms the robustness and applicability of this newly proposed method.
This study has connected aerosol optical properties with $N_{CCN}$, and also proposed a novel
method to calculate $N_{CCN}$ based only on measurements of a three-wavelength humidified
nephelometer system. Due to the simple operation and stability of the humidified nephelometer
system, this method will facilitate the real time monitoring of $N_{CCN}$, especially on aircrafts. In
addition, measurements of the widely used CCN counter are limited to supersaturations higher than
0.07. In fogs and shallow layer clouds, supersaturations are generally smaller than 0.1% (Ditas et al.,
2012; Hammer et al., 2014a, b; Krüger et al., 2014). For studying aerosol-cloud interaction, this
method is more applicable due to its applicability for calculating $N_{CCN}$ at lower supersaturations than

371     1.0%.

Acknowledgement
This work is supported by the National Natural Science Foundation of China (41590872 and

374     41505107).

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

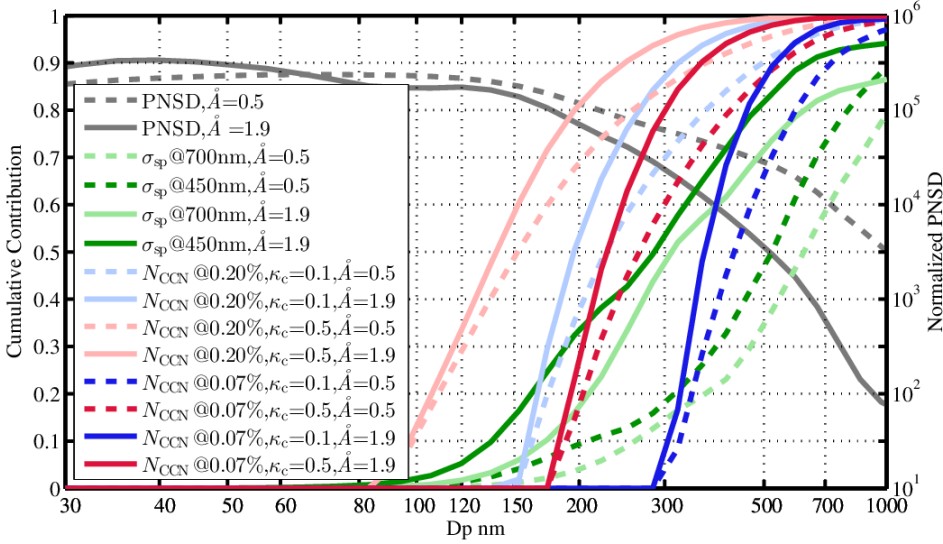


Figure 1.
Aerosol PNSD (black lines), the cumulative contribution of $\sigma_{sp}$ at wavelength of 450nm and 700nm
(dark green lines and light green lines, respectively), the cumulative contribution of $N_{CCN}$ at
supersaturation of 0.07% (dark red and dark blue lines) and the cumulative contribution of $N_{CCN}$ at
supersaturation of 0.20% (light red and light blue lines) based on measurement in several campaigns
in the North China Plain. Solid lines and dashed lines indicate Å of 1.9 and 0.5, respectively. Blue
lines and red lines indicate $\kappa_c$ of 0.1 and 0.5, respectively.

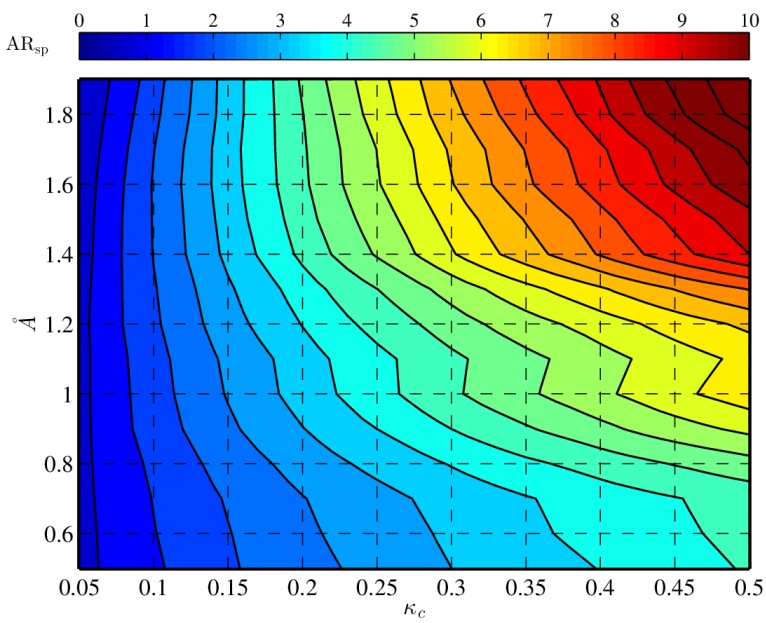


Figure 2.
Colors represent $AR_{sp}$ (calculated as $AR_{sp} = \frac{N_{CCN}}{\sigma_{sp}}$ at 450nm wavelength and 0.07% supersaturation)
with different PNSDs (classified by $\mathring{A}$ values) and different $\kappa_c$.

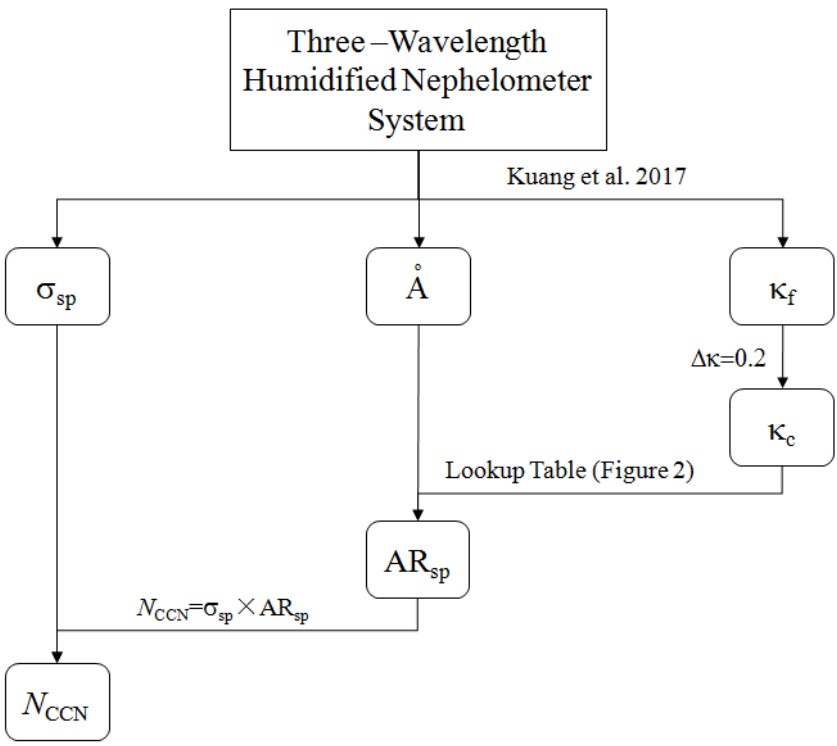


Figure 3.
The schematic chart of the $N_{\text{CCN}}$ prediction based on measurements of a humidified nephelometer
system.

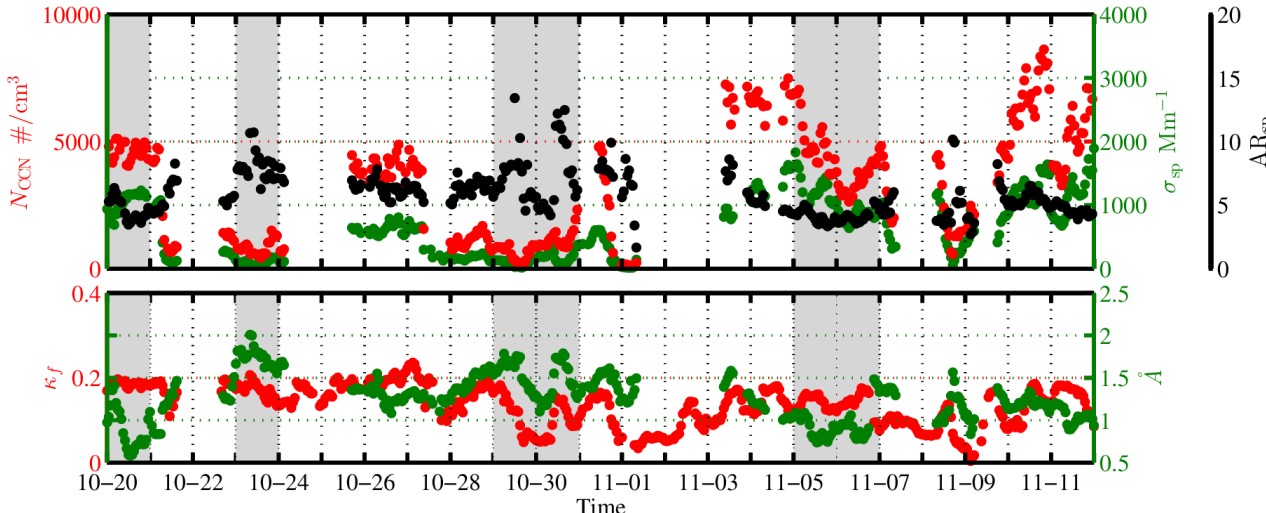


Figure 4.
Overview of measurements in Gucheng in 2016. Upper plot: time series of $N_{\text{CCN}}$ at the
supersaturation of 0.07% (red dots), $\sigma_{\text{sp}}$ at the wavelength of 50nm (green dots) and their ratios
(black dots), referred to as $\text{AR}_{\text{sp}}$. Lower plot: time series of $\kappa_{\text{f}}$ (red dots) and $\text{Å}$ (green dots). The
grey bars are periods when the sensitivity of $\text{AR}_{\text{sp}}$ to $\kappa_{\text{c}}$ is notable.

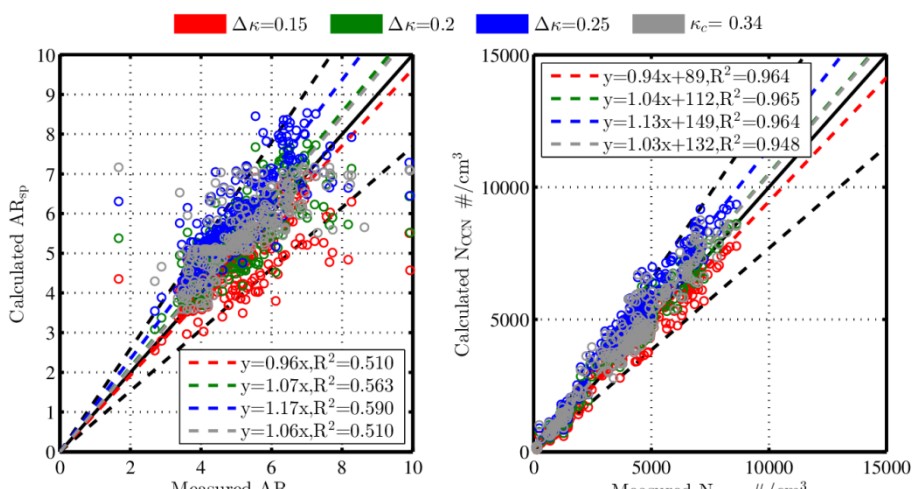


Figure 5.
Left plot: comparisons of calculated $AR_{sp}$ and measured $AR_{sp}$ with different conversions of $\kappa_c$ from
$\kappa_f$. Right plot: regressions of calculated $N_{CCN}$ and measured $N_{CCN}$ with different conversions of $\kappa_c$
from $\kappa_f$.

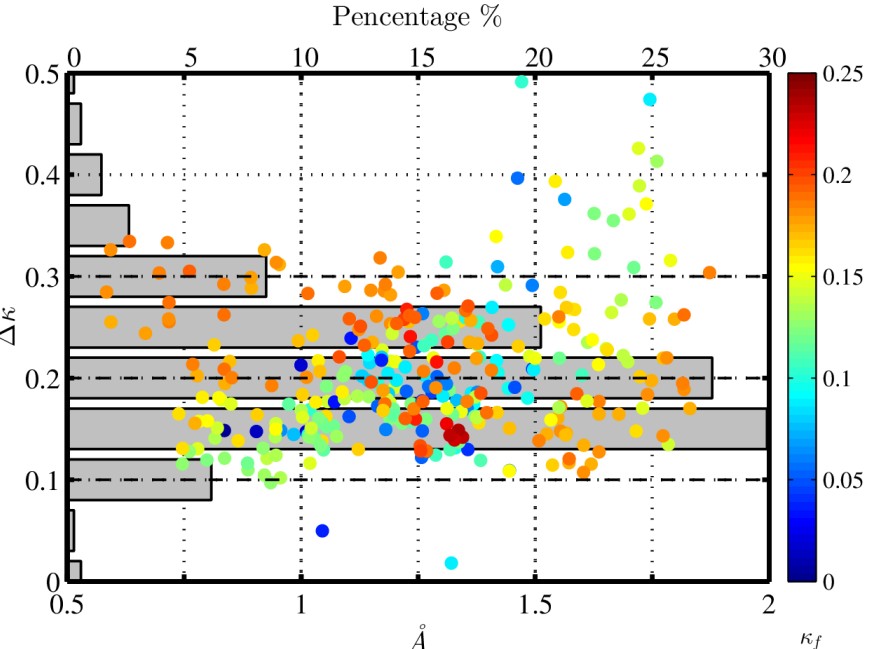


Figure 6.
Differences between $\kappa_c$ and $\kappa_f$, referred to as $\Delta\kappa$, with Å (positions of dots) and $\kappa_f$ (colors of
dots). Bars represent percentages of $\Delta\kappa$ within different ranges.


| Campaign | Air mass | Parameter | Caveats | Results | Reference |
|---|---|---|---|---|---|
| ICARTT[1] in the north eastern USA and Canada | Polluted air mass | fRH and PNSD | Calculate $N_{CCN}$ with aerosol hygroscopicity contrained by f(RH) and PNSD. | Predict $N_{CCN}$ at SS > 0.3% with a 0.9 $R^2$. | Ervens et al., 2007 |

| | | | | | |
|---|---|---|---|---|---|
| HaChi[2] on the North China Plain | Aged continental air mass | PNSD and fRH | Similar to Ervens et al., 2007. Calculate $N_{CCN}$ with the hygroscopicity parameter constrained by f(RH) and PNSD. | Slopes around 1 and $R^2$ around 0.9. | Chen et al., 2014 |
| TARFOX[3] Atlantic seaboard and ACE-2[4] | Polluted air mass | Retrieved aerosol volume from remote sensing | Predict $N_{CCN}$ from aerosol volumes with empirical number-to-volume concentration ratio | Overestimate up to 5 times | Gasso and Hegg, 2003 |
| ACE-2 in northeastern Atlantic | Diverse air mass | Backscatter or extinction profile. CCN at the surface. | Retrieve $N_{CCN}$ profile from backscatter (or extinction) vertical profile assuming their ratios are the same to the ratio at surface, which can be calculated by backscatter (or extinction) and $N_{CCN}$ measured at the surface | Predict $N_{CCN}$ on most days for 0.1% SS and on 20%–40% of the days at 1% SS. | Ghan and Collins, 2004 |
| ARM[5] Climate Research Facility central site at the Southern Great Plains | Continental air mass | Backscatter (or extinction) and RH profile. fRH and CCN at surface | Same as Ghan and Collins, 2004. | Explains CCN variance for 25%-63% of all measurements at high supersaturations | Ghan et al., 2006 |
| TRACE-P[6] and ACE-Asia[7] | Asian outflow over the western Pacific | Aerosol Index (AI, the product of ambient light extinction and Å) | Predict $N_{CCN}$ based on empirical relationship between AI and $N_{CCN}$ | AI relate well to CCN only with suitably stratified data | Kapustin et al., 2006 |
| Multiple measurements | Diverse air mass | AERONET aerosol optical thickness (AOT) | Predict $N_{CCN}$ based on empirical relationship between AOT and $N_{CCN}$ as a power law | Predict $N_{CCN}$ at SS > 0.3% with a 0.88 $R^2$, but have a factor-of-four range of $N_{CCN}$ at a given AOT | Andreae, 2009 |

| Four ARM sites | Polluted air mass | SSA, backscatter fraction and $\sigma_{sp}$ | Estimate $N_{CCN}$ from fitting parameters for the $N_{CCN}$ activity spectra, which can be calculate based on their empirical relationships with aerosol optical properties. | Predict $N_{CCN}$ with slopes around 0.9 and $R^2$ around 0.6. | Jefferson, 2010 |
|---|---|---|---|---|---|
| Multiple ARM sites around the world | Diverse air mass | RH, fRH, SSA, AOT and $\sigma_{sp}$ | Calculate $N_{CCN}$ with $\sigma_{sp}$ (or AOT) based on their empirical relationship, whose impact RH, fRH and SSA. | Achieve the best results by using $\sigma_{sp}$ and SSA. Weakly affect on the $\sigma_{sp}$–$N_{CCN}$ relationship by fRH. Deteriorate $N_{CCN}$–AOT relationship with increasing RH | Liu and Li, 2014 |
| Multiple ARM sites around the world | Diverse air mass not dominated by dust | Å and extinction coefficient | Calculate $N_{CCN}$ with light extinction based on their emperical relationship. | Deviate typically within a factor of 2.0. | Shinozuka et al., 2015 |

Table 1.

Review of studies that have used aerosol optical parameters to infer $N_{CCN}$.

[1] International Consortium for Atmospheric Research on Transport and Transformation.

[2] Haze in China.

[3] Troposphere Aerosol Radiative Forcing Experiment.

[4] Second Aerosol Characterization Experiment.

[5] Atmospheric Radiation Measurement.

[6] Transport and Chemical Evolution over the Pacific.

[7] Aerosol Characterization Experiment–Asia.