# Peer review of "A new method for calculating number concentrations of Cloud"

_Atmospheric Measurement Techniques, 2017_

## Referee Comment (RC1) · Anonymous Referee #1 · 19 Aug 2017

The authors present a study where they determined the number of cloud condensation nuclei (CCN) using a new method based on nephelometer measurements. They claim that this method is more convenient and cheaper than traditional measurements. Several studies have been published over the last 10 years that show that humidified nephelometer measurements can be used to infer CCN concentrations. They make several assumptions and use of various additional parameters. The apparent difference of the current study is the fact that no measurements of the particle number size distribution (PNSD). The manuscript contains several obscure sections and mistakes

(grammar, typos). In addition, the method is poorly described and compared to previous work. Several sections are not well organized. A major revision considering my detailed comments below might help to improve the manuscript such that it may be considered for publication. In addition, the complete manuscript should be carefully proofread.

Major comments 1) Applicability of the new method The caveats of the new method should be made clear in the abstract and conclusions. It is mentioned that it cannot be applied for externally mixed aerosol and particle populations with many large particles (e.g. dust, sea salt). Are there situations when delta(kappa) is too large/small that this bias will influence N(CCN)? Does the shape of the aerosol distribution play a role? Would, for example, multiple modes affect the Angstrom coefficient such that it exceeds 1.5?

2) Comparison to previous studies I suggest adding a table listing previous studies that have used optical aerosol parameters to infer N(CCN). This table should include the parameters that were used (PNSD etc), air mass characteristics (aged or not), caveats of the method and comments on results/findings. This way, the necessity of measurements for various air masses will be more obvious and the applicability of the new method will be clearer. For example, the difference to the methods by Kuang et al. and Brock et al. to the current method is not fully clear.

3) Clarity of method application a) While Figure 3 is somewhat helpful, it should be extended to be the central figure of the manuscript. Labels can be added to the arrows explaining in detail what is done in each step, e.g. a reference to the respective equation would be helpful.

b) The comparison to measured N(CCN) is useful and necessary in order to validate the new method. However, a few more details about the CCN measurements are needed. At what supersaturations were they measured (l. 116)? It is known that CCN measurements are most uncertain at low supersaturations. What supersaturation was

chosen for the comparison?

4) Clarity of language At several places, the text is not clear or even wrong and should be revised. Examples include:

l. 57: Aerosol hygroscopicity is defined as the ability of an aerosol particle to take up water. Hygroscopicity is not a function of particle size.

l. 68- 72: It should be clarified which combination of parameters is best suited and which problems/deviations (from what?) might occur.

l. 143: 'and can determines 'kappa' with A' is unclear

l. 174: This text is hard to follow. At the very least, add numerical ranges for the various parameters. It would be even better to connect this discussion to a figure (either an additional one or existing one)

l. 198: '. . .which reveals that particles. . .' – I do not understand this fragment

l. 214: Do you mean '..due to size-dependent hygroscopicity'?

l. 284 – 294: This paragraph should be rewritten as I cannot follow the line of thought. For example, you start with 'On one hand, the variation of kappa(c) can be quite large...' and continue later 'On the other hand, the influence of kappa(c) cannot be ignored. . .' These two sentences should introduce opposing facts, but they do not.

5) Structure Essential information should be given as early as possible in the manuscript:

a) The Angstrom coefficient should be defined in the introduction or in Section 2.

b) Caveats of the method should be pointed out throughout the paper

c) It is highly confusing that in Section 2 delta(kappa) is introduced as being 0.2 and only in Section 3 a lengthy discussion of this value is given and sensitivity studies are performed. A more thorough discussion of reasons and conditions of large or
small delta(kappa), respectively, should be added in the context of the applicability and accuracy of the new method. How would the results change if not a constant delta(kappa) but the exact difference for each data point in Fig 5 is used? Can we learn something from the resulting (dis)agreement as a function of A?

6) Formatting All parameters should be expressed in equations and should be formatted and numbered as such. For example, l. 101 and the definition of fRH (l. 106).

7) Figures

a) The caption of Figure 2 cannot be understood without reading the text. At the very least, the parameters should be spelled out and a reference to an equation in the text should be added.

b) What are the grey bars in Figure 4?

c) The grey symbols in Figure 6 overlap with many other symbols. Maybe choosing open symbols would improve clarity.

Minor comments

l. 64: Add references for the 'common use'.

l. 66: This sentence needs work: 1) word missing after 'carbonaceous'. 2) What is meant by 'most important hydrophobic'?

l. 135/6: S is not included in the equation

l. 164/5: Is this a result based on the literature or the current data set? If the former, add references.

l. 191: AR(sp) can only be 0 if N(CCN) or if sigma(sp) is infinitely large. Is either of this a realistic situation?

l. 245: What 'microphysical properties' are you referring to here? 'Composition' is a chemical property.

l. 247: 'more sensitive' as compared to which other parameter?

l. 249: Later and in Figure 2, the range of A is up to approx. 1.5, not 15

Technical comments

l. 2: 'Nuclei' misspelled

l. 94: an inlet . . . consisting of . . . an inline . . .

l. 109: AR has not been defined before.

l. 128: campaigns

l. 154: indicates

l. 159: wavelengths

l. 171: increases

l. 179: remove 'as'
* * *

---

## Referee Comment (RC2) · Anonymous Referee #2 · 29 Aug 2017

**Summary:**

This work proposed a new method to estimate number concentrations of CCN based on the humidified nephelometer measurements. The advantages of this method are more convenient and cheaper than traditional measurements, and no other measurements are needed. The manuscript fits well to the scope of AMT. Thus I recommend it to be published after the following comments listed below have been adequately addressed.

**Comments:**

1. Lines 47-52: Please add some texts to evaluate each application. Also, I agree with another reviewer that one table should be added to summary the previous studies using aerosol optical properties to calculate $N_{CCN}$.

2. Lines 172-176: I guess that the authors want to claim that the uncertainty will be smaller when performing this method for shorter wavelength and lower supsaturation. Am I correct? Concerning only one supersaturation (0.07%) was test in this study, and the relative deviation is within 30%. Therefore, I am wondering that is it possible to perform this method to higher supersaturations to check when the uncertainty will be larger than 50%.

3. Lines 180-181: How to calculate the differences (150 nm and 100 nm)? Please explain.

4. Line 191: What are smaller CCN-active particles? Do you mean Aitken mode particles? I think the contribution of particles smaller than 100 nm to $\sigma_{sp}$ is always negligible.

5. Lines 201-203: See comment 2. It seems that you claim 0.07% is the highest supersaturation that can be applied for this method. Why? Do you have results for other supersaturations?

6. Lines 206-208: Add references. Why do you think $\kappa_f$ is always lower than $\kappa_c$? Any explanations?

7. Lines 241-247 and Figure 5: How about the agreement between the retrieved and measured $\kappa_c$?

8. Lines 248-251: The authors claim that this method can only be adopted when Å is lower than 1.5. Is this conclusion only based on this study or can be used in different environments?

9. I suggest the authors reorganize or recheck the text for each figure caption. More information should be included, such as gray background in Figure 2 and black & dashed lines in Figure 6.

10. Technical comments:

Title: Nuclei.

Line 36: also.

Line 110: please provide DMA type.

Lines 111 and 120: an electrostatic classifier.

Line 126: campaigns.

Line 133: there is no S in Eq. (1), please reformulate it.

Line 137: explain $\kappa_f$.

Line 152: indicates.

Line 234: 0.5 to 1.5

Lines 271-273: please add references.

Line 308: changes

There are still several grammar mistakes in the text, please carefully check.

---

## Author Comment (AC1) · 1 Nov 2017

The responses and the revised manuscript are included in the supplement zip file.

Please also note the supplement to this comment:
https://www.atmos-meas-tech-discuss.net/amt-2017-193/amt-2017-193-AC1-supplement.zip

———————————————

---

## Author Response (AR1)

Dear Editor,

We greatly thank the reviewers for their detailed review. Responses addressing reviewers' comments point-by-point were uploaded (and also attached to this file). The manuscript has been revised and improved accordingly.

Best Regards

Chunsheng Zhao

**Response to Referee #1:**

**General comment:**

*The authors present a study where they determined the number of cloud condensation nuclei (CCN) using a new method based on nephelometer measurements. They claim that this method is more convenient and cheaper than traditional measurements. Several studies have been published over the last 10 years that show that humidified nephelometer measurements can be used to infer CCN concentrations. They make several assumptions and use of various additional parameters. The apparent difference of the current study is the fact that no measurements of the particle number size distribution (PNSD). The manuscr ipt contains several obscure sections and mistakes (grammar, typos). In addition, the method is poor ly described and compared to previous work. Several sections are not well organized. A major revision considering my detailed comments below might help to improve the manuscript such that it may be considered for publication. In addition, the complete manuscript should be carefully proofread.*

**Response:** Thanks for your comments. Comments are addressed point-by-point and corresponding responses are listed below. The whole manuscript is also checked.

**Major comments:**

*1) Applicability of the new method*

*The caveats of the new method should be made clear in the abstract and conclusions. It is mentioned that it cannot be applied for externally mixed aerosol and particle populations with many large particles (e.g. dust, sea salt). Are there situations when delta(kappa) is too large/small that this bias will influence N(CCN)? Does the shape of the aerosol distribution play a role? Would, for example, multiple modes affect the Angstrom coefficient such that it exceeds 1.5?*

**Response:** Thanks for the suggestion.

Hygroscopicity of both inorganic compounds and black carbon remain about the same under different saturated conditions, and the increase of hygroscopicity under supersaturated conditions are generally caused by organic compounds (Wex et al., 2009; Renbaum-Wolff et al., 2016). In ambient atmosphere, particles are consist of diverse compositions and the difference of particle hygroscopicity under different saturated conditions can be expected to be limited within a small range. However, in regions near strong sources of only one specific composition, this specific composition can completely dominate and lead to too large or too small $\Delta\kappa$. As aresult, in regions with strong sources of a single composition, $\Delta\kappa$ can be too large or too small and lead to significant deviations of predicte $N_{CCN}$.

Particle number size distribution (PNSD) is important for aerosol activation and aerosol scattering. In this study, both aerosol activation and aerosol scatterirng are considered to be dominated by particles of accumulation mode and the shape of PNSD is taken in account to some extent by Å ranging from 0.5 to 1.5. When aerosol populations consist of large number of particles of Atiken mode which can contribute significantly to aerosol scattering, Å can exceed 1.5 and $N_{CCN}$ predicted by this new method can be overestimated.

We have added corresponding descriptions in the abstract, results and conclusions as follows:

Abstract, Line 20: *"...is established to predict $N_{CCN}$. Due to the precondition for the application, this new method is not suitble for externally mixed particles, large particles (e.g. dust and sea salt) or particles near single source regions. ..."*

Results, the last line of the second-to-last paragraph: *"... In regions of single aerosol emissions or productions, the actual Δκ can be too large (some organic compositions, Wex et al., 2009; Renbaum-Wolff et al., 2016) or too small (inorganic compositions and black carbon) and thus is not suitble for the application of this method."*

Conclusions, the last line of second paragraph: *"... under conditions without sea salt aerosol, dust aerosol, externally mixed aerosol or aerosol near single source regions."*

*2) Comparison to previous studies*

*I suggest adding a table listing previous studies that have used optical aerosol parameters to infer N(CCN). This table should include the parameters that were used (PNSD etc), air mass characteristics (aged or not), caveats of the method and comments on results/findings. This way, the necessity of measurements for various air masses will be more obvious and the applicability of the new method will be clearer. For example, the difference to the methods by Kuang et al. and Brock et al. to the current method is not fully clear.*

**Response:** Thanks for the suggestion. We have added Table 1 and improved descriptions in the introduction. The methods proposed by Kuang et al. and Brock et al. are used to calculate hygroscopicity parameter and thus is not included in Table 1.

| Campaign | Air mass | Parameter | Caveats | Results | Reference |
|---|---|---|---|---|---|
| ICARTT[1] in the north eastern USA and Canada | Polluted air mass | fRH and PNSD | Calculate $N_{CCN}$ with aerosol hygrosopicity contrained by f(RH) and PNSD. | Predict $N_{CCN}$ at SS > 0.3% with a 0.9 $R^2$. | Ervens et al., 2007 |

| | | | | | |
|---|---|---|---|---|---|
| HaChi[2] on the North China Plain | Aged continental air mass | PNSD and fRH | Similar to Ervens et al., 2007. Calculate $N_{CCN}$ with the hygrosopicity parameter contrained by f(RH) and PNSD. | Slopes around 1 and $R^2$ around 0.9. | Chen et al., 2014 |
| TARFOX[3] Atlantic seaboard and ACE-2[4] | Polluted air mass | Retrieved aerosol volume from remote sensing | Predict $N_{CCN}$ from aerosol volumes with empirical number-to-volume concentration ratio | Overestimate up to 5 times | Gasso and Hegg, 2003 |
| ACE-2 in northeastern Atlantic | Diverse air mass | Backscatter or extinction profile. CCN at the surface. | Retrieve $N_{CCN}$ profile from backscatter (or extinction) vertical profile assuming their ratios are the same to the ratio at surface, which can be calculated by backscatter (or extinction) and $N_{CCN}$ measured at the surface | Predict $N_{CCN}$ on most days for 0.1% SS and on 20%–40% of the days at 1% SS. | Ghan and Collins, 2004 |
| ARM[5] Climate Research Facility central site at the Southern Great Plains | Continental air mass | Backscatter (or extinction) and RH profile. fRH and CCN at surface | Same as Ghan and Collins, 2004. | Explains CCN variance for 25%-63% of all measurements at high supersaturations | Ghan et al., 2006 |
| TRACE-P[6] and ACE-Asia[7] | Asian outflow over the western Pacific | Aerosol Index (AI, the product of ambient light extinction and Å) | Predict $N_{CCN}$ based on empirical relationship between AI and $N_{CCN}$ | AI relate well to CCN only with suitably stratified data | Kapustin et al., 2006 |
| Multiple measurements | Diverse air mass | AERONET aerosol | Predict $N_{CCN}$ based on empirical relationship | Predict $N_{CCN}$ at SS > 0.3% with a | Andreae, 2009 |

| | | | | | |
|---|---|---|---|---|---|
| | | optical thickness (AOT) | between AOT and $N_{CCN}$ as a power law | 0.88 $R^2$, but have a factor-of-four range of $N_{CCN}$ at a given AOT | |
| Four ARM sites | Polluted air mass | SSA, backscatter fraction and $\sigma_{sp}$ | Estimate $N_{CCN}$ from fitting parametes for the $N_{CCN}$ activity spectra, which can be calculate based on their emprical relationships with aerosol optical properties. | Predict $N_{CCN}$ with slopes around 0.9 and $R^2$ around 0.6. | Jefferson, 2010 |
| Multiple ARM sites around the world | Diverse air mass | RH, fRH, SSA, AOT and $\sigma_{sp}$ | Calculate $N_{CCN}$ with $\sigma_{sp}$ (or AOT) based on their empirical relationship, whose impact RH, fRH and SSA. | Achieve the best results by using $\sigma_{sp}$ and SSA. Weakly affect on the $\sigma_{sp}$–$N_{CCN}$ relationship by fRH. Deteriorate $N_{CCN}$–AOT relationship with increasing RH | Liu and Li, 2014 |
| Multiple ARM sites around the world | Diverse air mass not dominated by dust | Å and extinction coefficient | Calculate $N_{CCN}$ with light extinction based on their emperical relationship. | Deviate typically within a factor of 2.0. | Shinozuka et al., 2015 |

Tabel 1. Review of studies that have used aerosol optical parameters to infer $N_{CCN}$.

[1] International Consortium for Atmospheric Research on Transport and Transformation.

[2] Haze in China.

[3] Troposphere Aerosol Radiative Forcing Experiment.

[4] Second Aerosol Characterization Experiment.

[5] Atmospheric Radiation Measurement.

[6] Transport and Chemical Evolution over the Pacific.

[7] Aerosol Characterization Experiment–Asia.

*3)   Clarity of method application*

*a) While Figure 3 is somewhat helpful, it should be extended to be the central figure of the*
*manuscript. Labels can be added to the arrows explaining in detail what is done in each step,*
*e.g. a reference to the respective equation would be helpful.*

**Response:** Thanks for the suggestion. We have revised Figure 3 as follows:

[Figure]

*b) The comparison to measured N(CCN) is useful and necessary in order to validate the*
*new method. However, a few more details about the CCN measurements are needed. At what*
*supersaturations were they measured (l. 116)? It is known that CCN measurements are most*
*uncertain at low supersaturations. What supersaturation was chosen for the comparison?*

**Response:** Thanks for the suggestion. There were five supersaturations (0.07%, 0.10%,
0.20%, 0.40% and 0.80%) and $N_{CCN}$ at 0.07% supersaturation was chosen for the camparison,
because at higher supersaturations this new method is not applicable any more. We have
revised the statement in line 116 as: *"Measurements at five supersaturations (0.07%, 0.10%,*
*0.20%, 0.40% and 0.80%) were conducted sequentially with each cycle lasted for 1 hour, and*
*$N_{CCN}$ at 0.07% supersaturation was used in this study."*

*4)   Clarity of language*

*At several places, the text is not clear or even wrong and should be revised. Examples include:*

*l. 57: Aerosol hygroscopicity is defined as the ability of an aerosol particle to take up water. Hygroscopicity is not a function of particle size.*

**Response:** Thanks for the suggestion. We agree with the reivewer. The statement in the manuscript leads to misunderstanding and we have revised it as: *"···aerosol CCN activity is determined by aerosol size and aerosol hygroscopicity.···"*

*l. 68- 72: It should be clarified which combination of parameters is best suited and which problems/deviations (from what?) might occur.*

**Response:** Thanks for the suggestion. As mentioned in the response to "2) Comparison to previous studies", we have revised line 68-72 as follows:

*"...due to the diversity of hygroscopicity of less-absorbing components. Thus $N_{CCN}$ calculation combining SSA, backscatter fraction and $\sigma_{sp}$ still lead to significant deviations, with a 0.6 $R^2$ (Jefferson, 2010). As for fRH, there was a study applied aerosol optical quantities ($\sigma_{sp}$ or aerosol optical thickness) with fRH or SSA to calculate $N_{CCN}$ (Liu and Li, 2014). In their study, compared with the combination of SSA and aerosol optical quantities, the combination of fRH and aerosol optical quantities is found to be less effective in estimating $N_{CCN}$, even though fRH directly connected with aerosol hygroscopicity (Liu and Li, 2014). ..."*

*l. 143: 'and can determines 'kappa' with A' is unclear*

**Response:** Thanks for the suggestion. We have revised this sentence as: *"... and is found can be used to predict $\kappa_f$ in combination wih Å in recent studies(Brock et al., 2016;Kuang et al., 2017). This method of calculating $\kappa_f$ based on $\kappa_{sca}$ and Å was confirmed by good agreement with $\kappa_f$ calculated from fRH and PNSD."*

*l. 174: This text is hard to follow. At the very least, add numerical ranges for the various parameters. It would be even better to connect this discussion to a figure (either an additional one or existing one)*

**Response:** Thanks for the suggestion. We have revised Figure 1 as follows:

[Figure]

Figure 1. Aerosol PNSD (black lines), the cumulative contribution of $\sigma_{sp}$ at wavelength of 450nm and 700nm (dark green lines and light green lines, respectively), the cumulative contribution of $N_{CCN}$ at supersaturation of 0.07% (dark red and dark blue lines) and the cumulative contribution of $N_{CCN}$ at supersaturation of 0.20% (light red and light blue lines) based on measurement in several campaigns in the North China Plain. Solid lines and dashed lines indicate Å of 1.9 and 0.5, respectively. Blue lines and red lines indicate $\kappa_c$ of 0.1 and 0.5, respectively.

There is a typo error of Å value in Figure 1. And we also revised this sentence as *"Because particles smaller than 200nm can activate at supersaturations higher than 0.07% while scatter less light at wavelengths longer than 450nm, which are shown as the light color lines in Figure 1, ..."*

*l. 198: '...which reveals that particles...' – I do not understand this fragment*

**Response:** Thanks for the suggestion. We have revised this sentence as: *"This higher sensitivity of $AR_{sp}$ to Å reveals that particles having mean predominate size smaller than existing particles can contribute more to $N_{CCN}$."*

*l. 214: Do you mean '..due to size-dependent hygroscopicity'?*

**Response:** Thanks for the suggestion. We have revised it accordingly.

*l. 284 – 294: This paragraph should be rewritten as I cannot follow the line of thought. For example, you start with 'On one hand, the variation of kappa(c) can be quite large...' and continue later 'On the other hand, the influence of kappa(c) cannot be ignored . . .' These two sentences should introduce opposing facts, but they do not.*

**Response:** Thanks for the suggestion. We have revised this paragraph as follows:

*"... First, the variation of $\kappa_c$ is not always small and can cause non-ignorable deviations of calculated $N_{CCN}$ in certain cases. As many studies of $N_{CCN}$ measurement showed, the variation of $\kappa_c$ is often small and a constant $\kappa_c$ can be used to calculate $N_{CCN}$ accurately (Andreae and Rosenfeld, 2008;Gunthe et al., 2009;Rose et al., 2010;Deng et al., 2013). Results in this study are similar to these previous studies. However, large variations of $\kappa_c$ are also found in some other studies. In NCP, fluctuations of aerosol hygroscopicity during New Particle Formation events and soot emissions lead to significant deviations of calculated $N_{CCN}$ from average aerosol hygroscopicity (Ma et al., 2016). Second, the influence of $\kappa_c$ variation on $N_{CCN}$ calculation cannot be ignored because the value of the average hygroscopicity differs in various regions during various periods. In summer of NCP, measured $\kappa_f$ at sub-saturated conditions can reach up to 0.45 when inorganic compositions dominate in particles (Kuang et al., 2016). ..."*

*5) Structure*

*Essential information should be given as early as possible in the manuscript:*

*a) The Angstrom coefficient should be defined in the introduction or in Section 2.*

**Response:** Thanks for the suggestion. We have defined Angstrom Exponent in Section 2 and we have revised the statement in the introduction in line 47 as: "… Angstrom Exponent (Å, which is the exponent commonly used to describe the dependence of $\sigma_{sp}$ on wavelength),…"

*b) Caveats of the method should be pointed out throughout the paper*

**Response:** Thanks for the suggestion. We have added caveats in the abstract and conclusions as presented in lines 50-60 in this reponse.

*c) It is highly confusing that in Section 2 delta(kappa) is introduced as being 0.2 and only in Section 3 a lengthy discussion of this value is given and sensitivity studies are performed. A more thorough discussion of reasons and conditions of large or small delta(kappa), respectively, should be added in the context of the applicability and accuracy of the new method. How would the results change if not a constant delta(kappa) but the exact difference for each data point in Fig 5 is used? Can we learn something from the resulting (dis)agreement as a function of A?*

**Response:** Thanks for the suggestion.

We have added the discussion of reasons and conditions of $\Delta\kappa$ variations in the second to last paragraph in Section 3 as follows:

*"...a smaller difference of 0.1 between $\kappa_c$ and $\kappa_f$ should be used (Kuang et al., 2017). This simplified relationship between $\kappa_c$ and $\kappa_f$ is a rough estimate regardless of the complexity of differences of aerosol hygroscopicity measured by different instruments, but still used in this study for two reasons. First, the accurate conversion cannot be achieved without detailed information of the particle hygroscopicity, which is difficult and complicated to measure. Second, a deviation of $\kappa_c$ less than 0.1 generally leads to a deviation of $N_{CCN}$ less than 20% (Ma et al., 2016), which is comparable with the deviation of CCN measurements. As a result, for a simple method of $N_{CCN}$ calculation, this conversion is quite easy. In addition, it is important to note that the value of the difference between $\kappa_c$ and $\kappa_f$ is also a rough estimate regardless of the complexity of aerosol hygroscopicity under different conditions, and the influence of $\Delta\kappa$ deviation on $N_{CCN}$ calculation needs to be further examined based on field observation."*

The use of exact $\Delta\kappa$ for each data point to calculate $N_{CCN}$ means the application of measured $\kappa_c$ for $N_{CCN}$ prediction based on the lookup table in Figure 2. This exclude the uncertainty of aerosol hygroscopicity in predicting $N_{CCN}$ and highlight the impact of PNSD's variation on $N_{CCN}$ prediction when Å is used to estimate the influence of PNSD on the relationship between $N_{CCN}$ and $\sigma_{sp}$. Calculated $AR_{sp}$ and calculated $N_{CCN}$ with corresponding Å are shown in Figure S1. Relative deviations between calculated $AR_{sp}$ and measured $AR_{sp}$ are generally no higher than 30%. Compared with correlations shown in the left plot of Figure 6, whose correlation coefficients ranges from 0.5 to 0.6, the correlation between calculated $AR_{sp}$ based on measured $\kappa_c$ and measured $AR_{sp}$ is better, with a correlation coefficient of 0.709. As for calculated $N_{CCN}$ using measured $\kappa_c$, relative deviations are mainly within 30%. Deviations of calculated $AR_{sp}$ and calculated $N_{CCN}$ are due to variations of PNSDs which share a same Å. In addition, as for relative deviations of both calculated $AR_{sp}$ and calculated $N_{CCN}$, neither of them has a significant relationship with corresponding Å. Besides the uncertainty of CCN measurement, causes of calculated $N_{CCN}$ deviations also include
variations of PNSDs with a common Å are almost the same for different Å, showing random
fluctuations of PNSDs from their true values.

[Figure]

Figure S1. Left plot: comparisons of calculated $AR_{sp}$ based on measured $\kappa_c$ and measured
$AR_{sp}$. Right plot: regressions of calculated $N_{CCN}$ based on measured $\kappa_c$ and measured $N_{CCN}$.
The color of the dot are corresponding Å for each data point.

*6)  Formatting*

*All parameters should be expressed in equations and should be formatted and numbered*
*as such. For example, l. 101 and the definition of fRH (l. 106).*

**Response:** Thanks for the suggestion. We have revised them accordingly.

*7)  Figures*

*a) The caption of Figure 2 cannot be understood without reading the text. At the very least,*
*the parameters should be spelled out and a reference to an equation in the text should be*
*added.*

**Response:** Thanks for the suggestion. We have revised the caption as *"Colors represent $AR_{sp}$*
*(calculated as $AR_{sp} = \frac{N_{CCN}}{\sigma_{sp}}$ at 450nm wavelength and 0.07% supersaturation) with different*

*PNSDs (classified by $\text{Å}$ values) and different $\kappa_c$."*

*b) What are the grey bars in Figure 4?*

**Response:** Thanks for the suggestion. The grey bars are periods when the sensitivity of
$AR_{sp}$ to $\kappa_c$ is notable. We have added the description in the caption of Figure 4 and in the
third paragrph of Section 3.2.

*c) The grey symbols in Figure 6 over lap with many other symbols. Maybe choosing open*
*symbols would improve clarity*

**Response:** Thanks for the suggestion. We have revised Figure 6 as follows:

[Figure]

**Minor comments:**

*l. 64: Add references for the 'common use'.*

**Response:** Thanks for the suggestion. We have added references as follows:

Jefferson, A.: Empirical estimates of CCN from aerosol optical properties at four remote
sites, Atmos. Chem. Phys., 10, 6855-6861, 10.5194/acp-10-6855-2010, 2010.

Liu, J. J., and Li, Z. Q.: Estimation of cloud condensation nuclei concentration from
aerosol optical quantities: influential factors and uncertainties, Atmospheric Chemistry and
Physics, 14, 471-483, 10.5194/acp-14-471-2014, 2014.

*l. 66: This sentence needs work: 1) word missing after 'carbonaceous'. 2) What is meant by 'most important hydrophobic'?*

**Response:** Thanks for the suggestion. The statement in the manuscript is confusing and we have revised this sentence as: *"Black carbon dominates the absorption of solar radiation and is a main hydrophobic composition as well."*

*l. 135/6: S is not included in the equation*

**Response:** Thanks for the suggestion. It shoud be RH and we have revised it accordingly.

*l. 164/5: Is this a result based on the literature or the current data set? If the former, add references.*

**Response:** Thanks for the comment. It's based on literature and we have added references as follows:

Cheng, Y. F., A. Wiedensohler, et al. (2008). "Aerosol optical properties and related chemical apportionment at Xinken in Pearl River Delta of China." Atmospheric Environment 42(25): 6351-6372.

Ma, N., C. Zhao, et al. (2011). "Aerosol optical properties in the North China Plain during HaChi campaign: an in-situ optical closure study." Atmos. Chem. Phys 11(12): 5959-5973.

Liu, H. J., C. S. Zhao, et al. (2014). "Aerosol hygroscopicity derived from size-segregated chemical composition and its parameterization in the North China Plain." Atmos. Chem. Phys. 14(5): 2525-2539.

Kuang, Y., C. S. Zhao, et al. (2017). "A novel method for deriving the aerosol hygroscopicity parameter based only on measurements from a humidified nephelometer system." Atmospheric Chemistry and Physics 17(11): 6651-6662.

*l. 191: AR(sp) can only be 0 if N(CCN) or if sigma(sp) is infinitely large. Is either of this a realistic situation?*

**Response:** Thanks for the suggestion. $\sigma_{sp}$ wouldn't be infinitely large and $AR_{sp}$ should be higher than 0. We have revised it as: *"… and $AR_{sp}$ is higher than 0 and lower than 10. …"*

*l. 245: What 'microphysical properties' are you referring to here? 'Composition' is a chemical property*

**Response:** Thanks for the comment. We are referring to the shape of particle size
distribution and aerosol hygroscopicity, and we have revised the last two sentences in this
paragraph as: *"... rather than the shape of particle size distribution and aerosol*
*hygroscopicity. Variations of $AR_{sp}$ result from the variations of Å and $\kappa_c$, which indicate the*
*variations of aerosol microphysical properties and chemical compositions."*

*l. 247: 'more sensitive' as compared to which other parameter?*

**Response:** Thanks for the comment. It should be $\kappa_c$ and we have revised it accordingly.

*l. 249: Later and in Figure 2, the range of A is up to approx. 1.5, not 15*

**Response:** Thanks for the suggestion. We have revised it accordingly.

**Technical comments**

*l. 2: 'Nuclei' misspelled*

**Response:** Thanks for the suggestion. We have revised it.

*l. 94: an inlet . . . consisting of . . . an inline . . .*

**Response:** Thanks for the suggestion. We have revised them.

*l. 109: AR has not been defined before.*

**Response:** Thanks for the suggestion. We have revised it.

*l. 128: campaigns*

**Response:** Thanks for the suggestion. We have revised it.

*l. 154: indicates*

**Response:** Thanks for the suggestion. We have revised it.

*l. 159: wavelengths*

**Response:** Thanks for the suggestion. We have revised it.

*l. 171: increases*

**Response:** Thanks for the suggestion. We have revised it.

*l. 179: remove 'as'*

**Response:** Thanks for the suggestion. We have revised it.

**Response to Referee #2:**

**General comment:**

*This work proposed a new method to estimate number concentrations of CCN based on the humidified nephelometer measurements. The advantages of this method are more convenient and cheaper than traditional measurements, and no other measurements are needed. The manuscript fits well to the scope of AMT. Thus I recommend it to be published after the following comments listed below have been adequately addressed.*

**Response:** Thanks for the comments. Comments are addressed point-by-point and corresponding responses are listed below.

**Specific comments:**

*1. Lines 47-52: Please add some texts to evaluate each application. Also, I agree with another reviewer that one table should be added to summary the previous studies using aerosol optical properties to calculate NCCN.*

**Response:** Thanks for the suggestion. We have added descriptions and a table as follows:

*"... Compared with the first kind, whose $R^2$ can be about 0.9, instruments used in the second kind of methods are cheaper and easier in operation, but has a lower accuracy of $R^2$ much lower than 0.9. …"*

| Campaign | Air mass | Parameter | Caveats | Results | Reference |
|---|---|---|---|---|---|
| ICARTT[1] in the north eastern USA and Canada | Polluted air mass | fRH and PNSD | Calculate $N_{CCN}$ with aerosol hygrosopicity contrained by f(RH) and PNSD. | Predict $N_{CCN}$ at SS > 0.3% with a 0.9 $R^2$. | Ervens et al., 2007 |
| HaChi[2] on the North China Plain | Aged continental air mass | PNSD and fRH | Similar to Ervens et al., 2007. Calculate $N_{CCN}$ with the hygrosopicity parameter contrained by f(RH) and PNSD. | Slopes around 1 and $R^2$ around 0.9. | Chen et al., 2014 |
| TARFOX[3] Atlantic seaboard and | Polluted air mass | Retrieved aerosol volume from | Predict $N_{CCN}$ from aerosol volumes with empirical number-to-volume | Overestimate up to 5 times | Gasso and Hegg, 2003 |

| | | | | | |
|---|---|---|---|---|---|
| ACE-2[4] | | remote sensing | concentration ratio | | |
| ACE-2 in northeastern Atlantic | Diverse air mass | Backscatter or extinction profile. CCN at the surface. | Retrieve $N_{CCN}$ profile from backscatter (or extinction) vertical profile assuming their ratios are the same to the ratio at surface, which can be calculated by backscatter (or extinction) and $N_{CCN}$ measured at the surface | Predict $N_{CCN}$ on most days for 0.1% SS and on 20%–40% of the days at 1% SS. | Ghan and Collins, 2004 |
| ARM[5] Climate Research Facility central site at the Southern Great Plains | Continental air mass | Backscatter (or extinction) and RH profile. fRH and CCN at surface | Same as Ghan and Collins, 2004. | Explains CCN variance for 25%-63% of all measurements at high supersaturations | Ghan et al., 2006 |
| TRACE-P[6] and ACE-Asia[7] | Asian outflow over the western Pacific | Aerosol Index (AI, the product of ambient light extinction and Å) | Predict $N_{CCN}$ based on empirical relationship between AI and $N_{CCN}$ | AI relate well to CCN only with suitably stratified data | Kapustin et al., 2006 |
| Multiple measurements | Diverse air mass | AERONET aerosol optical thickness (AOT) | Predict $N_{CCN}$ based on empirical relationship between AOT and $N_{CCN}$ as a power law | Predict $N_{CCN}$ at SS > 0.3% with a 0.88 $R^2$, but have a factor-of-four range of $N_{CCN}$ at a given AOT | Andreae, 2009 |
| Four ARM sites | Polluted air mass | SSA, backscatter fraction and $\sigma_{sp}$ | Estimate $N_{CCN}$ from fitting parametes for the $N_{CCN}$ activity spectra, which can be calculate based on their emperical relationships with aerosol optical properties. | Predict $N_{CCN}$ with slopes around 0.9 and $R^2$ around 0.6. | Jefferson, 2010 |
| Multiple ARM sites | Diverse air mass | RH, fRH, SSA, AOT | Calculate $N_{CCN}$ with $\sigma_{sp}$ (or AOT) based on their | Achieve the best results by using $\sigma_{sp}$ | Liu and Li, 2014 |

| | | and $\sigma_{sp}$ | empirical relationship, whose impact RH, fRH and SSA. | and SSA. Weakly affect on the $\sigma_{sp}$–$N_{CCN}$ relationship by fRH. Deteriorate $N_{CCN}$–AOT relationship with increasing RH | |
|---|---|---|---|---|---|
| around the world | | | | | |
| Multiple ARM sites around the world | Diverse air mass not dominated by dust | Å and extinction coefficient | Calculate $N_{CCN}$ with light extinction based on their emperical relationship. | Deviate typically within a factor of 2.0. | Shinozuka et al., 2015 |

Tabel 1. Review of studies that have used aerosol optical parameters to infer $N_{CCN}$.

[1] International Consortium for Atmospheric Research on Transport and Transformation.

[2] Haze in China.

[3] Troposphere Aerosol Radiative Forcing Experiment.

[4] Second Aerosol Characterization Experiment.

[5] Atmospheric Radiation Measurement.

[6] Transport and Chemical Evolution over the Pacific.

[7] Aerosol Characterization Experiment–Asia.

*2. Lines 172-176: I guess that the authors want to claim that the uncertainty will be smaller when*
*performing this method for shorter wavelength and lower supsaturation. Am I correct?*
*Concerning only one supersaturation (0.07%) was test in this study, and the relative deviation is*
*within 30%. Therefore, I am wondering that is it possible to perform this method to higher*
*supersaturations to check when the uncertainty will be larger than 50%.*

**Response:** Thanks for the suggestion. Yes, the uncertainty is smaller when performing this
method for shorter wavelength and lower supersaturation. We apply this method to higher
supersaturations and compare calculated $AR_{sp}$ with measured $AR_{sp}$. $\Delta\kappa$ at five supersaturations are
all set to be 0.2. Results are shown in Figure S2 as follows:

[Figure]

Figure S2. (a) to (e) Calculated $AR_{sp}$ (ratios between $N_{CCN}$ and $\sigma_{sp}$, represented as the color) based on $\sigma_{sp}$ and $N_{CCN}$ with different PNSDs classified by Å and different $\kappa_c$ at the five supersaturations. (f) Comparison between calculated $AR_{sp}$ and measured $AR_{sp}$. Colors represent supersaturations.

As the lookup table at each supersaturation shown, calculated $AR_{sp}$ is higher at higher supersaturation as a whole, which indicate more CCN with a common $\sigma_{sp}$. The same as shown in Figure 6, relative deviations of calculated $AR_{sp}$ from measured $AR_{sp}$ are generally within 30%. Calculated $AR_{sp}$ at 0.10% supersaturation are 30% higher than measured $AR_{sp}$ but still associated with measured $AR_{sp}$. For the three supersaturations higher than 0.10%, relative deviations of calculated $AR_{sp}$ from measured $AR_{sp}$ often exceed 50% and there is no significant correlation between calculated $AR_{sp}$ and measured $AR_{sp}$. Results shown in Figure S2 verify the conclusion that the uncertainty is smaller when performing this method for shorter wavelength and lower supersaturation and the this method is not applicable at supersaturations higher than 0.10%.

*3. Lines 180-181: How to calculate the differences (150 nm and 100 nm)? Please explain.*

**Response:** Thanks for the suggestion. The diameter difference of cumulative contribution between 0.5 Å and 1.7 Å is roughly estimated by the average of differences where cumulative contributions range from 0.2 to 0.8. We have revised the statement as: *"In detail, differences of cumulative contribution curves between 0.5 Å and 1.7 Å are about 150nm for $\sigma_{sp}$ and about 100nm for $N_{CCN}$, by estimating the average of differences of diameters where cumulative contributions range from 0.2 to 0.8"*

4. *Line 191: What are smaller CCN-active particles? Do you mean Aitken mode particles? I think the contribution of particles smaller than 100 nm to σsp is always negligible.*

   **Response:** Thanks for the comment. Smaller CCN-active particles refers to particles smaller than the avergae diameter of the whole CCN-active particles but is still CCN-active. For example, paticles with diameters slightly larger than $D_c$ contribute less to $\sigma_{sp}$ than paticles with diameters much larger than $D_c$. We have added the corresponding description after the sentence.

5. *Lines 201-203: See comment 2. It seems that you claim 0.07% is the highest supersaturation that can be applied for this method. Why? Do you have results for other supersaturations?*

   **Response:** Thanks for the suggestion. Yes, as for the five supersaturations measured in this study, 0.07% is the highest supersaturation (also the only supersaturation) that can be applied for this method. This is because $N_{CCN}$ at supersaturations higher than 0.07% are dominated by small paticles more significantly than $\sigma_{sp}$ (shown in Figure 1) and the correlation between $N_{CCN}$ and $\sigma_{sp}$ become weaker. The result in Figure S2 shows that relative deviations of calculated $N_{CCN}$ at supersaturations higher than 0.07 can exceed 30% commonly.

6. *Lines 206-208: Add references. Why do you think κf is always lower than κc? Any explanations?*

   **Response:** Thanks for the suggestion. We have added reference as follows:

   *Irwin, M., N. Good, et al. (2010). "Reconciliation of measurements of hygroscopic growth and critical supersaturation of aerosol particles in central Germany." Atmos. Chem. Phys. 10(23): 11737-11752.*

   *Good, N., D. O. Topping, et al. (2010). "Consistency between parameterisations of aerosol hygroscopicity and CCN activity during the RHaMBLe discovery cruise." Atmospheric Chemistry and Physics 10(7): 3189-3203.*

   *Wex, H., M. D. Petters, et al. (2009). "Towards closing the gap between hygroscopic growth and activation for secondary organic aerosol: Part 1-Evidence from measurements." Atmospheric*

*Chemistry and Physics 9(12): 3987-3997.*

*Renbaum-Wolff, L., M. Song, et al. (2016). "Observations and implications of liquid–liquid phase separation at high relative humidities in secondary organic material produced by α-pinene ozonolysis without inorganic salts." Atmos. Chem. Phys. 16(12): 7969-7979.*

There are mainly two reasons why $\kappa_f$ is always lower than $\kappa_c$. First, $\kappa_f$ is calculated base on measurments under subsaturated conditions while $\kappa_c$ is calculated base on measurments under supersaturated conditions. Studies found that aerosol hygroscopicity can increase under supersaturated conditions, due to dissolution of slightly soluble substances (Wex et al., 2009), the phase separation of organic compounds (Renbaum-Wolff et al., 2016) and so on. Second, accumulation mode paticles are generally most hygroscopic. $\kappa_f$ represent the average hygroscopticiy of total particles and is generally lower than hygroscopicities of accumulation mode paticles (Kuang et al., 2017), while $\kappa_c$ at 0.07% supersaturation indicate hygroscopicities of particles around 200nm. As a result, $\kappa_f$ is always lower than $\kappa_c$.

*7.  Lines 241-247 and Figure 5: How about the agreement between the retrieved and measured κc?*

**Response:** Thanks for the comment. We compare the retrieved and the measured $\kappa_c$, as shown in Figure S3. For the majority of data points, relative deviations between retrieved and measured $\kappa_c$ are within about 20%. A large relative deviation much higher than 20% usually correspond to a Å higher than 1.5, which is also shown in Figure 5.

[Figure]

Figure S3. Comparisons between measured and retrieved $\kappa_c$ (dots) and their corresponding Å values (colors). The solid black line is the 1:1 line and dashed black lines indicate relative deviations of 20%.

8. *Lines 248-251: The authors claim that this method can only be adopted when Å is lower than 1.5. Is this conclusion only based on this study or can be used in different environments?*

**Response:** Thanks for the suggestion. This conclusion can be used in other environment, however the lookup table should be recalculated based on PNSD measured in corresponding environment.

9. *I suggest the authors reorganize or recheck the text for each figure caption. More information should be included, such as gray background in Figure 2 and black & dashed lines in Figure 6.*

**Response:** Thanks for the suggestion. There is no gray background in Figure 2 while gray backgrounds in Figure 4 are not described yet. We have revised them accordingly.

10. *Technical comments:*

*Title: Nuclei.*

**Response:** Thanks for the suggestion. We have revised it.

*Line 36: also.*

**Response:** Thanks for the suggestion.

*Line 110: please provide DMA type.*

**Response:** Thanks for the suggestion. It's Model 3081 DMA and we have revised it.

*Lines 111 and 120: an electrostatic classifier.*

**Response:** Thanks for the suggestion. We have revised them.

*Line 126: campaigns. Line 133: there is no S in Eq. (1), please reformulate it.*

**Response:** Thanks for the suggestion. It shoud be relative humidity (RH) and we have revised it.

*Line 137: explain $\kappa f$.*

**Response:** Thanks for the suggestion.

*Line 152: indicates.*

**Response:** Thanks for the suggestion. We have revised it.

*Line 234: 0.5 to 1.5*

**Response:** Thanks for the suggestion. We have revised it.

*Lines 271-273: please add references.*

**Response:** Thanks for the suggestion.

*Line 308: changes*

**Response:** Thanks for the suggestion. We have revised it.

*There are still several grammar mistakes in the text, please carefully check.*

**Response:** Thanks for the suggestion.

**Reference:**

Wex, H., M. D. Petters, et al. (2009). "Towards closing the gap between hygroscopic growth and activation for secondary organic aerosol: Part 1-Evidence from measurements." Atmospheric Chemistry and Physics 9(12): 3987-3997.

[revised manuscript text omitted]

---

## Author Response (AR2)

Dear Editor,

We greatly thank the reviewers for their detailed review. Responses addressing reviewers' comments point-by-point were uploaded (and also attached to this file). The manuscript has been revised and improved accordingly.

Best Regards

Chunsheng Zhao

**Response to Referee #1:**

**General comment:**

*The authors have improved the manuscript and clarified some unclear sections. However, not all of my comments have been addressed in the manuscript; in addition, some obscurities remain and new ones were introduced by new text.*

*The language has been improved somewhat. As I expect that copyediting of the manuscript will take care of it, I only listed a few wording or grammar mistakes below.*

*Line numbers in my comments refer to the marked-up version of the revised manuscript, attached to the response to the reviews.*

**Response:** Thanks for your comments. We have addressed comments point-by-point and list corresponding responses below. We also have checked the language, and corrected wording and grammar mistakes.

**Major comments:**

*1) Limitations*

*The authors do a better job now pointing out the caveats of the new method. However, one limitation is the restriction to S = 0.07% (or lower). However, some more discussion should be given considering the following:*

*-In my previous comments, I had pointed out that CCN counters are least accurate at low S. This uncertainty in measurements should be mentioned.*

**Response:** Thanks for the suggestion. We have added this description in section 2.1 as follows:

*"Due to non-idealities of CCN counter at supersaturations lower than 0.10%, CCN measurement at 0.07% supersaturation was found to be the most uncertain (Rose et al., 2008) and can lead to deviations of measured $N_{CCN}$ in this study."*

*- In addition, the authors mention at the very end of the manuscript that the low S makes this method 'more applicable for ambient measurements of clouds and fogs in the atmosphere'. Typical supersaturations in clouds range from < 0.1% (e.g. for stratus) to > 1% (for cumulus clouds). This should be mentioned and appropriate references added.*

**Response:** Thanks for the suggestion. We have added references and revised the last sentence at the end of the manuscript as follows:

*"In fogs and shallow layer clouds, supersaturations are generally smaller than 0.1% (Ditas et al.,*

*2012; Hammer et al., 2014a, b; Krüger et al., 2014). For studying aerosol-cloud interaction, this method is more applicable due to its applicability for calculating $N_{CCN}$ at lower supersaturations than 1.0%."*

*- What is the reasoning that you can assume internal mixing of the aerosol? (e.g. l. 264)*

**Response:** Thanks for the comment. The reason why internal mixing of the aerosol can be assumed is that the deviation of $N_{CCN}$ calculation due to this assumption is generally small. This small deviation results from the more significant influence of aerosol size and aerosol hygroscopicity than aerosol mixing state in determining aerosol CCN activity (Dusek et al., 2006; Ervens et al., 2010). In the new method of this study, influences of aerosol size and aerosol hygroscopicity on $N_{CCN}$ calculation are considered by introducing Angstrom Exponent and kappa, respectively. Using Angstrom Exponent and kappa instead of aerosol size and aerosol hygroscopicity increases the deviation of $N_{CCN}$ calculation, which can be much larger than the deviation due to the assumption of aerosol mixing state. Thus the improvement of $N_{CCN}$ calculation by using a more detailed assumption of aerosol mixing state than internal mixing is little in this new method. We have added corresponding description in the last paragraph in section 3.1 as follows:

*"… In addition, it should be noted that influences of aerosol hygroscopicity and aerosol size on aerosol CCN activity are more significant than aerosol mixing state and the deviation of $N_{CCN}$ calculation due to the assumption of aerosol mixing state is smaller than the deviation due to aerosol size and aerosol hygroscopicity. In the new method of this paper, using Å and $\kappa_c$ to indicate the influence of aerosol size and aerosol hygroscopicity on aerosol CCN activity will increase the deviation of $N_{CCN}$ calculation, which is much larger than the deviation due to the assumption of aerosol mixing state. As a result, the improvement of $N_{CCN}$ calculation by introducing a more detailed mixing state than internal mixing is limited and aerosol populations can be assumed to be internally mixed for simplification. Thus this method …"*

*- N(CCN) is rarely measured at cloud height as surface measurements are much simpler. It always remains the question whether surface aerosol is actually connected to clouds above the measurement site. However, I do not believe that just the transport time of bringing aerosol aloft (can be as little as a few minutes) is a sufficient ageing time - as implied in l. 268 - to achieve internal mixing. Please support or reject this assumption by appropriate references.*

**Response:** Thanks for the suggestion. Yes, surface aerosol is not always connected with aerosol within clouds and the transport time of bringing aerosol aloft is generally shorter than that needs to achieve internal mixing. Aerosol at cloud height can be aged and internally mixed when there is weak vertical transport or transport from upwind regions. The assumption of internal mixing state is found to be generally reliable by several studies (McMeeking et al., 2011; Ferrero et al., 2014). Thus the new method proposed in this study is generally applicable for measurement at cloud forming height. We have revised the statement around line 268 as follows:

*"... For regions above the boundary layer where clouds form and measurements of $N_{CCN}$ are important, aerosol generally tends to be internally mixed when there is no strong vertical transport (McMeeking et al., 2011; Ferrero et al., 2014) and no plumes ... In summary, this method can be used to calculate $N_{CCN}$ for air mass tending to be dominated by aged aerosol particles like continental regions and clouds forming heights."*

*2) Structure*

*The discussion of delta(kappa) is still poorly organized. I suggest starting with the calculated delta(kappa) and its sensitivity studies and then concluding that an assumption of 0.2 is sufficiently good. That way, Section 3 will be better organized and also the conclusions could be structured better.*

**Response:** Thanks for the suggestion. We have reorganized the discussion of delta kappa in section 3. Before the discussion, we reviewed Gucheng campaign and examined the new method for $N_{CCN}$ calculation based on Gucheng data. Then we demonstrated the calculated delta kappa, studied the sensitivity of calculated NCCN to delta kappa and drew the conclusion at last. In this way, sequences of Figure 5 and Figure 6 are exchanged with each other and corresponding paragraphs are adjusted. The paragraph of the discussion is shown as follows:

*"In addition, the variation of $\Delta\kappa$ and its influence on $AR_{sp}$ and $N_{CCN}$ calculation are studied . As shown in Figure 6, $\Delta\kappa$ is around 0.2 and independent from Å and $\kappa_c$ and over 80% of $\Delta\kappa$ ranges from 0.1 to 0.3. A notable deviation of $\Delta\kappa$ can only be found when Å is higher than 1.5. High values of Å represent existence of small particles, which tend to be fresh emitted and experience inefficient aging processes. In this case, this simplified conversion of $\kappa_c$ may not be applicable. Furthermore, $\Delta\kappa$ with different values are applied in the new method to calculate $N_{CCN}$. In the first way, $\Delta\kappa$ of the $\kappa_c$ conversion is set to be 0.05 higher or lower, which means $\Delta\kappa$ of 0.25 or 0.15. The corresponding results are presented as the red dots and blue dots in Figure 5. In the second way, a constant $\kappa_c$ of 0.34, which is the average of $\kappa_c$ values in Gucheng campaign, is used to calculate $AR_{sp}$ and $N_{CCN}$, and shown as the grey dots in Figure 5. In general, differences among calculations using various $\kappa_c$ conversions are quite small. The $\Delta\kappa$ difference of 0.05 in $\kappa_c$ conversion only leads to a difference of 10% for the system relative deviation of calculated $N_{CCN}$. The correlation*

*coefficient of the calculation using a constant $\kappa_c$ is just a little lower than correlation coefficients of*

*calculations using a $\kappa_c$ conversion. As a result, for data measured in Gucheng campaign, the*

*method of calculating $N_{CCN}$ is insensitive to the uncertainty of the $\kappa_c$ conversion and a $\Delta\kappa$ of 0.2 is*

*applicable in this new method."*

We have also revised the third paragraph in conclusions as follows:

*"... $AR_{sp}$ is around 5 and changes with Å and $\kappa_f$. Based on this new method, $N_{CCN}$ are*

*calculated to compare with its measured values. The agreement between the calculated $N_{CCN}$ and the*

*measured $N_{CCN}$ is achieved with relative deviations less than 30%. Furthermore, the variation of $\Delta\kappa$*

*and its influence on $N_{CCN}$ calculation are studied. The difference between $\kappa_f$ and $\kappa_c$, was $0.2\pm0.1$.*

*Sensitivity of calculated $N_{CCN}$ ..."*

**Minor comments:**

*l. 21 and later in the manuscript: I don't understand why it is restricted to 'single source regions'.*
*All what matters is whether the aerosol is aged or fresh – whereas the latter could originate from*
*multiple emission sources.*

**Response:** Thanks for the comment. We have revised "*particles near single source regions*" as
"*fresh aerosol particles*". And we also revised "*aerosol near single source regions*" in the last
sentence of the second paragraph in conclusions as "*fresh aerosol*".

*l. 30: 'nuclei' is plural; either 'nuclei are… ' or 'nucleus is…'*

**Response:** Thanks for the suggestion. We have revised "is" as "are".

*l. 49: Why not adding the equation of the Angstrom exponent here (and remove it later)?*

**Response:** Thanks for the suggestion. We have added the equation and removed it in the section
2.1.

*l. 51 and 52: What does R^2 refer to? In order to make it easier to read, I suggest splitting this*
*sentence into (at least) two. That way it would also become clear what' has' in l. 52 refers to.*

**Response:** Thanks for the suggestion. We have revised this sentence as follows:

*"Coefficient of determination ($R^2$) between measured and calculated $N_{CCN}$ using the first kind of*

*method is about 0.9. For the second kind of method, $R^2$ is generally lower than 0.9, although the used instruments are cheaper and easier in operation."*

*l. 62: As mentioned in my previous comments, aerosol hygroscopicity is defined as the ability of a particle (or a material in general) to take up a certain amount of water at a given RH. It is NOT a function of aerosol size as the text here suggests.*

**Response:** Thanks for the comment. Yes, aerosol hygroscopicity is not a function of aerosol size and the text is misleading. We have revised it as follows:

*"... aerosol CCN activity is determined by aerosol size and aerosol chemical composition, and aerosol chemical composition can be defined as aerosol hygroscopicity. ⋯"*

*l. 68 and l. 108: Define parameters only once. Here you use two different names for rRH*

**Response:** Thanks for the suggestion. We have removed the second definition and added the equation of fRH in the first place.

*l. 71 and remainder of the manuscript: 'Composition' is often used wrongly. 'Component' is the right word*

**Response:** Thanks for the suggestion. We have revised them as follows:

*"hydrophobic composition" to "hydrophobic components", "organic compositions" to "organic components", "inorganic compositions" to "inorganic components" and "hygroscopic compositions" to "hygroscopic components".*

*l. 172: Do you really refer to Eq.-1 here?*

**Response:** Thanks for the comment. It should be Eq. 3 and we have revised it accordingly. In addition, we have corrected numbers of equations.

*l. 221: I still don't understand this sentence: what are 'particles' as opposed to 'existing particles'?*

**Response:** Thanks for the comment. This statement is confusing and we have revised this sentence as follows:

*"This higher sensitivity of $AR_{sp}$ to Å reveals that, if the mean predominate size of particles is*

*smaller, the increase of $N_{CCN}$ due to the increase of Å mentioned in the former paragraph can be larger as a result."*

*l. 253-255: How is 'too large' or 'too small' defined?*

**Response:** Thanks for the comment. A 'too large' delta kappa can be about 4 times of kappa values and a 'too small' delta kappa can be zero. We have revised the sentence as follows:

*"... the actual $\Delta\kappa$ can be too large (about 4 times of kappa values for some organic compositions, ...) or too small (nearly zero for inorganic compositions and black carbon..."*

*l. 315: This is a very strong statement. Is this true under all conditions?*

**Response:** Thanks for the comment. No, it's applicable for data measured in Gucheng campaign. We have revised it accordingly.

*l. 318 and l. 325: In my previous comments, I had pointed out that 'on the one hand' and 'on the other hand' are used to introduce two opposing statements. However, here they introduce pretty much the same thing, i.e. the variation of kappa(c) can be quite large vs the influence of kappa(c) cannot be ignored. This should be restructured.*

**Response:** Thanks for the comment. We have revised "on the one hand" and "on the other hand" into "however" and "furthermore", respectively.

*l. 335: A 'cloud chamber' is not the same as a CCN counter. In the former, a cloud is formed and the supersaturation cannot be exactly predetermined and/or measured. In a CCN counter, particles are exposed to a preset supersaturation. I assume you mean the latter here.*

**Response:** Thanks for the suggestion. Yes, it should be the CCN counter. And we have revised it accordingly.

**Technical comments**

*l. 19: involves --> includes*

**Response:** Thanks for the suggestion. We have revised it accordingly.

*l. 21: suitable*

**Response:** Thanks for the suggestion. We have revised it accordingly.

*l. 77: '0.6 R^2' seems very colloquial. Better is 'R^2 = 0.6'.*

**Response:** Thanks for the suggestion. We have revised it accordingly.

*l. 77: leads*

**Response:** Thanks for the suggestion. We have revised it accordingly.

*l. 78: ..study that applied…*

**Response:** Thanks for the suggestion. We have revised it accordingly.

*l. 81: 'accurate' might be better to use than 'effective'*

**Response:** Thanks for the suggestion. We have revised it accordingly.

*l. 82: rRH is directly connected*

**Response:** Thanks for the suggestion. We have revised it accordingly.

*l. 159: Reword '..is found can be used'*

**Response:** Thanks for the suggestion. We have revised it as "… can be used …".

*l. 167: Define Dc here.*

**Response:** Thanks for the suggestion. We have revised it accordingly.

*l. 203-205: This sentence needs to be rewritten as it is confusing: How can NCCN range to 100 nm (it is in [cm-3]); what are the units of a 'cumulative contribution' if it ranges from 0.2 to 0.8?*

**Response:** Thanks for the comment. This sentence is confusing and we have revised it as follows:

*"In detail, cumulative contribution curves of $\sigma_{sp}$ at 1.9 Å is about 0.3 higher than curves at 0.5 Å at the size range of 200nm to 700nm. While cumulative contribution curves of $N_{CCN}$ at 1.9 Å is no higher than 0.2 higher than curves at 0.5 Å."*

*l. 217, 218: particles*

**Response:** Thanks for the suggestion. We have revised it accordingly.

Reference

[revised manuscript text omitted]
[3] | Polluted air | Retrieved | Predict $N_{CCN}$ from aerosol | Overestimate up to 5 | Gasso and |

| | | | | | |
|---|---|---|---|---|---|
| Atlantic seaboard and ACE-2[4] | mass | aerosol volume from remote sensing | volumes with empirical number-to-volume concentration ratio | times | Hegg, 2003 |
| ACE-2 in northeastern Atlantic | Diverse air mass | Backscatter or extinction profile. CCN at the surface. | Retrieve $N_{CCN}$ profile from backscatter (or extinction) vertical profile assuming their ratios are the same to the ratio at surface, which can be calculated by backscatter (or extinction) and $N_{CCN}$ measured at the surface | Predict $N_{CCN}$ on most days for 0.1% SS and on 20%–40% of the days at 1% SS. | Ghan and Collins, 2004 |
| ARM[5] Climate Research Facility central site at the Southern Great Plains | Continental air mass | Backscatter (or extinction) and RH profile. fRH and CCN at surface | Same as Ghan and Collins, 2004. | Explains CCN variance for 25%-63% of all measurements at high supersaturations | Ghan et al., 2006 |
| TRACE-P[6] and ACE-Asia[7] | Asian outflow over the western Pacific | Aerosol Index (AI, the product of ambient light extinction and Å) | Predict $N_{CCN}$ based on empirical relationship between AI and $N_{CCN}$ | AI relate well to CCN only with suitably stratified data | Kapustin et al., 2006 |
| Multiple measurements | Diverse air mass | AERONET aerosol optical thickness (AOT) | Predict $N_{CCN}$ based on empirical relationship between AOT and $N_{CCN}$ as a power law | Predict $N_{CCN}$ at SS > 0.3% with a 0.88 $R^2$, but have a factor-of-four range of $N_{CCN}$ at a given AOT | Andreae, 2009 |
| Four ARM sites | Polluted air mass | SSA, backscatter fraction and $\sigma_{sp}$ | Estimate $N_{CCN}$ from fitting parameters for the $N_{CCN}$ activity spectra, which can be calculate based on their empirical relationships with aerosol optical properties. | Predict $N_{CCN}$ with slopes around 0.9 and $R^2$ around 0.6. | Jefferson, 2010 |

| | | | | | |
|---|---|---|---|---|---|
| Multiple ARM sites around the world | Diverse air mass | RH, fRH, SSA, AOT and $\sigma_{sp}$ | Calculate $N_{CCN}$ with $\sigma_{sp}$ (or AOT) based on their empirical relationship, whose impact RH, fRH and SSA. | Achieve the best results by using $\sigma_{sp}$ and SSA. Weakly affect on the $\sigma_{sp}$–$N_{CCN}$ relationship by fRH. Deteriorate $N_{CCN}$–AOT relationship with increasing RH | Liu and Li, 2014 |
| Multiple ARM sites around the world | Diverse air mass not dominated by dust | Å and extinction coefficient | Calculate $N_{CCN}$ with light extinction based on their emperical relationship. | Deviate typically within a factor of 2.0. | Shinozuka et al., 2015 |

Table 1.

Review of studies that have used aerosol optical parameters to infer $N_{CCN}$.

[1] International Consortium for Atmospheric Research on Transport and Transformation.

[2] Haze in China.

[3] Troposphere Aerosol Radiative Forcing Experiment.

[4] Second Aerosol Characterization Experiment.

[5] Atmospheric Radiation Measurement.

[6] Transport and Chemical Evolution over the Pacific.

[7] Aerosol Characterization Experiment–Asia.